# The Miocene primate *Pliobates* is a pliopithecoid

Florian Bouchet [1] ✉, Clément Zanolli [2], Alessandro Urciuoli [1,3,4,5], Sergio Almécija [1,6,7], Josep Fortuny [1], Josep M. Robles[1], Amélie Beaudet [8,9,10], Salvador Moyà-Solà [1,11,12] & David M. Alba [1] ✉

The systematic status of the small-bodied catarrhine primate *Pliobates cataloniae*, from the Miocene (11.6 Ma) of Spain, is controversial because it displays a mosaic of primitive and derived features compared with extant hominoids (apes and humans). Cladistic analyses have recovered *Pliobates* as either a stem hominoid or as a pliopithecoid stem catarrhine (i.e., preceding the cercopithecoid–hominoid divergence). Here, we describe additional dental remains of *P. cataloniae* from another locality that display unambiguous synapomorphies of crouzeliid pliopithecoids. Our cladistic analyses support a close phylogenetic link with poorly-known small crouzeliids from Europe based on (cranio)dental characters but recover pliopithecoids as stem hominoids when postcranial characters are included. We conclude that *Pliobates* is a derived stem catarrhine that shows postcranial convergences with modern apes in the elbow and wrist joints—thus clarifying pliopithecoid evolution and illustrating the plausibility of independent acquisition of postcranial similarities between hylobatids and hominids.

The small-bodied catarrhine primate *Pliobates cataloniae*, from the Miocene (11.6 Ma) of northeast Spain, was originally described on the basis of a partial skeleton and associated cranium[1]. The systematic position of *Pliobates* among catarrhine primates (i.e., Old World monkeys, apes, and humans) remains unsettled. Dental similarities with dendropithecids from the Miocene of East Africa, interpreted as stem catarrhines[2,3]—i.e., preceding the divergence between cercopithecoids (Old World Monkeys) and hominoids (apes and humans)— or as basal hominoids[1,4,5], were noted[1]. However, *Pliobates* was originally interpreted as a stem hominoid more derived than proconsulids based on the results of a cladistic analysis (Fig. 1)[1]. Given the mosaic of both cranial and postcranial primitive (stem catarrhine-like) and derived (crown catarrhine and, especially, hominoid-like) features displayed by *Pliobates*, other authors proposed that this taxon might be alternatively interpreted as a pliopithecoid (i.e., a stem catarrhine) postcranially convergent with hominoids[6,7]. The latter view has been supported by subsequent cladistic analyses[2,3,8] (Fig. 1). A recent analysis of *Pliobates* carotid canal anatomy showed no particular similarities with either the pliopithecoid *Epipliopithecus* or hylobatids[9] and preliminary morphometric analyses of the inner ear semicircular

[1]Institut Català de Paleontologia Miquel Crusafont (ICP-CERCA), Universitat Autònoma de Barcelona, 08193 Cerdanyola del Vallès, Barcelona, Spain. [2]Univ. Bordeaux, CNRS, MCC, PACEA, UMR 5199, F-33600 Pessac, France. [3]Universitat Autònoma de Barcelona, Campus de la UAB, 08193 Cerdanyola del Vallès, Barcelona, Spain. [4]Division of Palaeoanthropology, Senckenberg Research Institute and Natural History Museum Frankfurt, Frankfurt am Main, Germany. [5]Universidad de Alcalá, Cátedra de Otoacústica Evolutiva y Paleoantropología (HM Hospitales-UAH), Departamento de Ciencias de la Vida, 28871 Alcalá de Henares, Madrid, Spain. [6]Division of Anthropology, American Museum of Natural History, New York, NY 10024, USA. [7]New York Consortium in Evolutionary Primatology, New York, NY 10016, USA. [8]Laboratoire de Paléontologie, Évolution, Paléoécosystèmes et Paléoprimatologie (PALEVOPRIM), UMR 7262 CNRS, Univ. Poitiers, Poitiers, France. [9]Department of Archaeology, University of Cambridge, Cambridge CB2 1QH, United Kingdom. [10]School of Geography, Archaeology, and Environmental Studies, University of the Witwatersrand, Johannesburg, WITS 2050, South Africa. [11]Institució Catalana de Recerca i Estudis Avançats, 08010 Barcelona, Spain. [12]Unitat d'Antropologia Biològica (Departament de Biologia Animal, de Biologia Vegetal i d'Ecologia), Universitat Autònoma de Barcelona, 08193 Cerdanyola del Vallès, Barcelona, Spain. ✉e-mail: florian.bouchet@icp.cat; david.alba@icp.cat

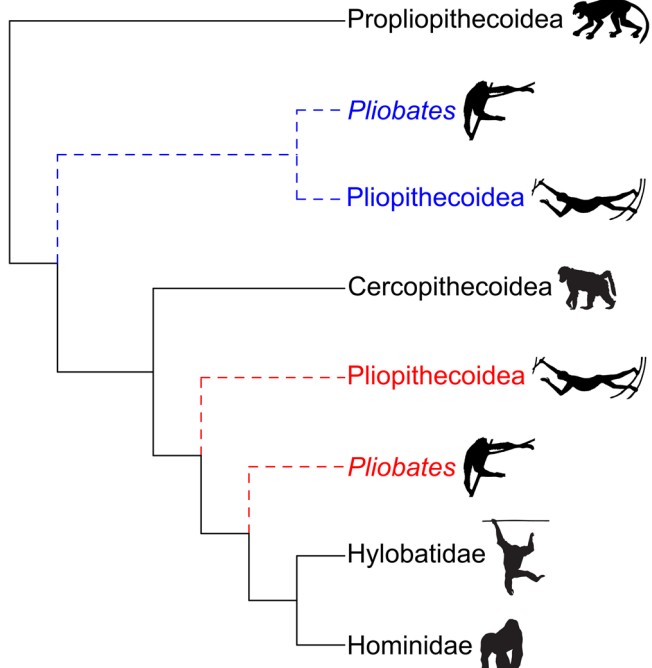

**Fig. 1 | Simplified cladogram of extant and extinct catarrhine primates depicting the two main phylogenetic hypotheses previously supported for *Pliobates*.** The red dashed lines depict the original interpretation of pliopithecoids and *Pliobates* as successive stem hominoids[1], whereas the blue dashed lines denote the alternative hypothesis of *Pliobates* being a stem catarrhine more closely related to pliopithecoids[2]. Black silhouettes (not to scale) of Propliopithecoidea, Cercopithecoidea, Hylobatidae, and Hominidae were taken from the following sources: Propliopithecoidea from Fig. 3 in ref. 71; Cercopithecoidea, Hylobatidae, and Hominidae from Summary Figure in ref. 13, reprinted with permission from AAAS; black silhouette (not to scale) of Pliopithecoidea was redrawn from Fig. 15.12 in ref. 72, reprinted with permission from Elsevier.

canals supported a stem catarrhine status more derived than propliopithecoids and *Epipliopithecus*, despite some similarities with the latter[10].

The contradictory results yielded by previous cladistic analysis for *Pliobates*—i.e., stem hominoid[1] vs. pliopithecoid[2,3] largely stem from the different emphasis put by the respective authors on postcranial features. The original topology recovering *Pliobates* as a hominoid was largely driven by the possession of numerous purported crown hominoid postcranial synapomorphies, which many authors consider homoplastic (i.e., independently evolved) to a large extent among various extant ape lineages[11–15]. Given the impossibility of disentangling homologous from homoplastic features a priori, an alternative approach to test whether *Pliobates* is a pliopithecoid would be to focus on dental morphology—as it is on this basis that pliopithecoids are distinguished from other catarrhines and considered to constitute a clade[8,16–20]. The holotype of *P. cataloniae* only preserves rather worn upper cheek teeth that did not enable ascertaining the most diagnostic features of pliopithecoids in the original description[1]. In contrast, here we describe unpublished, more diagnostic dentognathic material of *P. cataloniae* that includes the lower molars and is key to settling the debate about its systematic position, allowing us to confirm its pliopithecoid status. This material comes from Abocador de Can Mata (ACM) locality ACM/C5-D1, which is spatially close to and roughly coeval with ACM/C8-A4[21,22], the type locality of *P. cataloniae*. Based on the new material, we provide an emended diagnosis of this species and further evaluate its phylogenetic relationships among pliopithecoids by means of a cladistic analysis at the species level based on a taxon-character data matrix that includes qualitative and quantitative dental characters. To further evaluate the phylogenetic relationships

of *P. cataloniae* among catarrhines and the phylogenetic signal provided by other anatomical areas, we also conducted additional cladistic analyses at the genus level, including a wider representation of extinct and extant hominoids and considering (cranio)dental and postcranial characters both jointly and separately. Our results indicate that *Pliobates* is most closely related to crouzeliids, a diverse and widely distributed pliopithecoid family that was thus far mostly known from dental remains. They further support that *Pliobates* is not closely related to crown hominoids despite the possession of remarkable postcranial similarities, which has deep implications for our current understanding of ape and human evolution that are explored in this paper.

## Results
### Systematic paleontology

Order Primates Linnaeus, 1758
Suborder Anthropoidea Mivart, 1864
Infraorder Catarrhini É. Geoffroy Saint-Hilaire, 1812
Superfamily Pliopithecoidea Zapfe, 1961
Family Crouzeliidae Ginsburg and Mein, 1980
Subfamily Crouzeliinae Ginsburg and Mein, 1980
Genus *Pliobates* Alba et al. 2015
*Pliobates cataloniae* Alba et al. 2015

**Holotype.** IPS58443, a partial skeleton with associated skull of an adult female individual (see Table 1 in ref. 1 for details). It includes two dentognathic specimens: IPS58443.1, a right maxillary fragment with the I1–C1 alveoli, partial P3, and P4–M3 series; and IPS58443.2, a left maxillary fragment with M2 (partial) and M3. See Table 1 for dental measurements.

**Referred specimens.** Four maxillary and mandibular fragments and 12 isolated teeth from locality ACM/C5-D1 (see Supplementary Table 1 for a list of specimens). See Table 1 for dental measurements.

**Type locality.** ACM/C8-A4 (ACM composite stratigraphic sequence, els Hostalets de Pierola, Vallès-Penedès Basin, NE Iberian Peninsula).

**Distribution and age.** Only known from two localities of the ACM sequence dated to 11.6 Ma, thus roughly coinciding with the Middle/Late Miocene boundary[21,22]; see Materials & Methods for further details.

**Emended diagnosis.** (Diagnosis based on dental features emended from the original description; for cranial and postcranial features, see ref. 1; see Supplementary Text 1 for a differential diagnosis; see Supplementary Figs. 1–3 for comparison plates). Small-bodied catarrhine (~5 kg) with adult dental formula 2.1.2.3. Molars with buccolingually compressed cusps and sharp crests. Upper dental arcade somewhat divergent, with slightly heteromorphic upper incisors. I1 spatulate, mesiodistally waisted at the cervix, and very tilted mesialward. C1 moderately compressed and larger in males. Upper cheek teeth with extensive distal basin (premolars) or trigon basin (molars). Upper premolars ovoid with heteromorphic cusps, protocone more mesial and peripheral than paracone, distinct lingual cingulum (in the P4 only), and postparacrista forming an abrupt angle with the distal marginal ridge. Upper molars moderately (M1–M2) to very (M3) broad relative to length, with M2 larger than M1 and M3; small paraconule; small metacone and reduced hypocone (even more so in M3); protocone more distal than paracone; C-shaped lingual cingulum that does not surround the hypocone; and discontinuous buccal cingulum. M1–M2 with well-developed lingual cingulum and peripheral hypocone with prehypocrista directed toward the protocone but also a

secondary crest linking it to the crista obliqua. M3 markedly trapezoidal with rudimentary distal cusps. Lower incisors waisted toward cervix, i2 with a distinct distal prong. Female c1 with a marked cuspule-like thickening where the mesiolingual cristid meets the marked lingual cingulid. p3 with a single main cuspid. p4 suboval, with heteromorphic and mesially located metaconid and protoconid, rudimentary distal cuspids, poorly-developed buccal cingulid, and extensive talonid basin. Lower molars (m1–m2) with five main cuspids; protoconid more mesial than metaconid; hypoconulid centrally located; hypoprotocristid emerging distally from the protoconid; moderately inclined cristid obliqua (more so in the m1); extensive mesial fovea and larger talonid basin with incomplete pliopithecine triangle (well-developed distal arm) and not completely isolated from the distal fovea; broad and continuous buccal cingulid. DI1 similar to I1, with a moderately ledge-like lingual cingulum. DP3 trapezoidal with indistinct metacone and no crista obliqua. DP4 larger than DP3 and similar to M1–M2 but more triangular and with a comparatively smaller hypocone. dp3 mesially very tapering, with protoconid much more mesially located than metaconid and a lingually open mesial fovea (no distinct premetacristid). dp4 similar to m1 but more mesially tapering, with a longer mesial fovea, a more obliquely oriented cristid obliqua (directed toward the hypoprotocristid and hypometacristid junction, so that the postprotocristid originates from the latter junction), no pliopithecine triangle, and a discontinuous buccal cingulid.

## Description of the referred specimens

The previously unpublished specimens of *P. cataloniae* from ACM/C5-D1 include four dentognathic fragments and 12 isolated teeth (Supplementary Table 1). Descriptions of specimen preservation and occlusal shape are provided in Supplementary Texts 2 and 3, respectively. Although upper cheek tooth morphology was described based on the holotype[1], some of the new specimens display a lesser degree of wear and thus can be used to refine the assessment of the occlusal morphology of *P. cataloniae*. These remains from ACM/C5-D1 provide the unique opportunity to describe several tooth loci previously unknown for this taxon: I1, i2, female c1, p4, m1–m2, DI1, and both upper and lower deciduous premolars (Figs. 2–4). Most informative is the female infant mandible IPS43936 (Figs. 2g–i and 3), which enables to ascertain the morphology of the deciduous lower premolars (dp3–dp4) and digitally extract the germs of the permanent teeth, including the lower molars (m1–m2) (Figs. 2x, 4k, l and o, p). 3D models of the OES and EDJ of the 11 scanned specimens are presented in Fig. 4.

The occlusal morphology in the best preserved maxillary fragment from ACM/C5-D1 is comparable to that of the holotype (Fig. 1 in ref. 1; Fig. 2a–c). The difference in size between I1 and I2 alveoli confirms some degree of incisor heteromorphy. The isolated I1 shows that the crown is moderately low and labiolingually broader than mesiodistally long, spatulate, mesiodistally waisted at the cervix, and tilted mesially with an inclined apical margin (Fig. 2z). The C1 alveolus in the maxillary fragment is much larger than that of the holotype and most likely belongs to a male individual, further suggesting a marked degree of canine sexual dimorphism. The P3 provides further evidence of the morphology of this tooth (Figs. 2n and 4c), which was damaged in the holotype. The P4s (including a germ; Figs. 2n and 4d, e) display in turn an entirely comparable morphology to that of the holotype. Both upper premolars are broader than long and rather ovoid, with the two main cusps that are somewhat buccolingually compressed and heteromorphic, with the protocone being more peripheral than the paracone. The single hypoparacrista separates the restricted mesial fovea from the extensive and deep distal basin, while the postparacrista and the distal marginal ridge constitute an abrupt angle. The buccal cingulum is variably developed but always discontinuous whereas the lingual cingulum, only well-developed in the P4 (remnant in the P3), is short but wide. The six additional M1s and M2s from ACM/C5-D1 (Figs. 2n–r and 4f–i) are similar in occlusal morphology to one

another and to the corresponding teeth in the holotype, but additional details can be ascertained due to the less advanced degree of wear. They display a moderately broad subsquare occlusal contour that is somewhat buccolingually waisted at crown midlength with four main cusps, which are buccolingually compressed and connected by sharp crests. The protocone is more extensive than the paracone and metacone, which are similar in size in M1 but not in M2 (due to the smaller metacone), and slightly more distally situated than the paracone. The hypocone is smaller than the trigon cusps and more peripheral than the protocone. A small paraconule is present at the end of the preprotocrista. The crista obliqua is distinct and continuous. The mesial fovea is restricted and buccally located, while the trigon basin is much more extensive and deeper and the distal fovea is intermediate in size. Next to the prehypocrista, which is directed toward the protocone, there is a secondary crest that links the hypocone with the crista obliqua (this cannot be ascertained in 4 out of 9 molars due to wear). The buccal cingulum is discontinuous, whereas the lingual cingulum is broad (shelf-like) and C-shaped but does not surround the hypocone. When the whole sample is considered, the M1s and M2s display some variability in the degree of cusp compression, crest sharpness, lingual cingulum development, and hypocone size. However, the two first features largely depend on the degree of wear, with unworn or slightly worn specimens evincing more compressed cusps and sharper crests than those with a more advanced degree of wear.

**Table 1 | Dental measurements of *Pliobates cataloniae***

| Catalog No. | Tooth | Sex | MD | BLm | BLd | BLI |
|---|---|---|---|---|---|---|
| IPS42977 | L DP4 | ? | 5.5 | 5.7 | 5.4 | 103.6 |
| IPS42977 | L P4 | ? | (5.0) | (6.4) | | (128.0) |
| IPS43013 | L DP3 | ? | 3.9 | 4.8 | | 123.1 |
| IPS43433 | L c1 | F | 4.6 | 3.5 | | 76.1 |
| IPS43488 | L I1 | ? | >3.9 | (3.6) | | – |
| IPS43758 | R M1 | ? | 5.0 | 6.1 | 6.0 | 122.0 |
| IPS43758 | L M2 | ? | (5.8) | 6.8 | 6.4 | (117.2) |
| IPS43820 | R p4 | ? | >3.9 | >4.7 | | – |
| IPS43936 | L dp3 | ? | >3.9 | 3.1 | | – |
| IPS43936 | L dp4 | ? | 5.0 | 3.5 | 3.8 | 76.0 |
| IPS43936 | L m1 | ? | (6.1) | (4.4) | (4.7) | (77.0) |
| IPS43936 | L m2 | ? | (6.6) | (4.9) | (5.0) | (75.8) |
| IPS44014 | L P3 | M | 4.2 | 6.1 | | 145.2 |
| IPS44014 | L P4 | M | 4.1 | 5.9 | | 143.9 |
| IPS44014 | L M1 | M | 5.3 | 6.4 | 6.1 | 120.8 |
| IPS44273 | L i2 | ? | 3.0 | 3.8 | | 126.7 |
| IPS44393 | R dI1 | ? | 3.3 | 2.5 | | 75.8 |
| IPS58443.1 | R P3 | F | >2.8 | >4.5 | | – |
| IPS58443.1 | R P4 | F | 3.2 | 5.3 | | 165.6 |
| IPS58443.1 | R M1 | F | 5.0 | 5.9 | 5.8 | 118.0 |
| IPS58443.1 | R M2 | F | 5.3 | 6.5 | 6.4 | 122.6 |
| IPS58443.1 | R M3 | F | 4.6 | 6.4 | 5 | 139.1 |
| IPS58443.2 | L M2 | F | >5.0 | – | – | – |
| IPS58443.2 | L M3 | F | 4.5 | 6.7 | 5.1 | 148.9 |
| IPS93524 | R p4 | ? | >3.9 | >4.1 | | – |
| IPS94888 | L M2 | ? | 5.5 | 7.0 | 6.6 | 127.3 |
| IPS100379 | R M1 | ? | (5.2) | (6.1) | (5.7) | (117.3) |
| IPS100384 | L M2 | ? | 6.1 | 7.5 | 7.0 | 123.0 |
| IPS106878 | L dp4 | ? | 4.7 | 3.2 | 3.4 | 72.3 |

Values within parentheses are estimated measurements whereas the 'greater than' symbol (>) denotes incomplete measurements due to damage.

*BLI* breadth/length index, *BLd* buccolingual breadth at distal lobe, *BLm* buccolingual breadth at mesial lobe, *F* female, *M* male, *MD* mesiodistal length, *?* unknown sex.

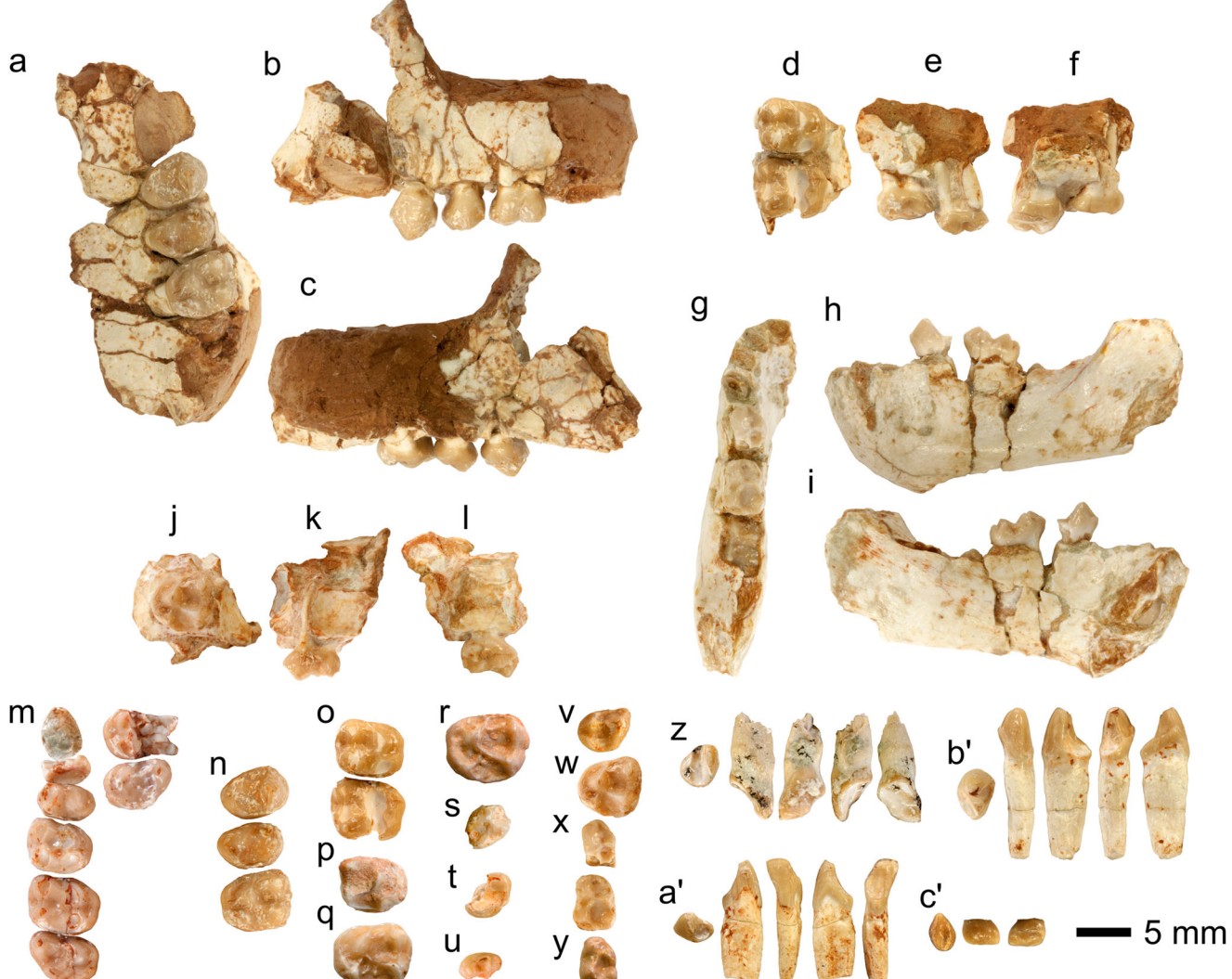

**Fig. 2 | Dentognathic fragments and teeth of *Pliobates cataloniae*.** Cheek teeth are depicted in occlusal view; for other specimens, views are indicated. **a–c** left male maxillary fragment with P3–M1 (IPS44014), in occlusal (**a**), buccal (**b**), and lingual (**c**) views; **d–f** right maxillary fragment with M1–M2 (IPS43758), in occlusal (**d**), buccal (**e**), and lingual (**f**) views; **g–i** left female mandibular fragment with dp3–dp4 and m1 inside crypt (IPS43936), in occlusal (**g**), buccal (**h**), and lingual (**i**) views; **j–l** left maxillary fragment with dP4 (IPS42977), in occlusal (**j**), buccal (**k**), and lingual (**l**) views; **m** right C1 alveolus and P3–M3 series (P3 broken) and left M2–M3 series (M2 broken) (IPS58443.1, holotype, female); **n** left P3–M1 series (IPS44014, male); **o** right M1–M2 series (IPS403758); **p** right M1 (IPS100379); **q** left M2 germ (IPS94888); **r** left M2 germ (IPS100384); **s** damaged right p4 (IPS93524); **t** partial right p4 (IPS43820); **u** right m2 germ mesial fragment (IPS94886); **v** left dP3 (IPS43013); **w** left dP4 (IPS42977); **x** left dp3–dp4 series (dp3 damaged) (IPS43936, female); **y** left dp4 (IPS106878); **z** left I1 (IPS43488), in occlusal, mesial, labial, distal, and lingual views; **a'** left i2 (IPS44273), in occlusal, mesial, labial, distal, and lingual views; **b'** left female c1 (IPS43433), in occlusal, mesial, labial, distal, and lingual views; **c'** right dI1 germ (IPS44393), in occlusal, labial, and lingual views. In occlusal view, mesial is on top. All the remains are from ACM/C5-D1 except the holotype, which is from ACM/C8-A4.

The lower incisor germs from the mandible (Fig. 3c, d) include the almost complete i1 crown and the still unfinished i2 crown. Coupled with the isolated i2 (Fig. 2a'), they show that both incisors are moderately high-crowned but that the i2 crown is much more asymmetrical, being mesially tilted and displaying a distinct distal prong at about crown midheight. The female c1 (Fig. 2b' and 3e) is low-crowned and displays three cristids that originate from the apex of the single cuspid, the distobuccal cristid being longer than the mesiolingual and distolingual cristids; the mesiolingual cristid forms a marked cuspule-like thickening at about crown midheight when it merges with the narrow lingual cingulid. The incompletely formed p3 germ of IPS43936 (Fig. 3f) shows that there is a single main cuspid (the protoconid) from which three cristids originate. The p4s, including an incompletely formed germ (Figs. 2s, t, 3g, and 4j), denote a suboval occlusal contour with well-developed albeit heteromorphic mesial cuspids (the metaconid being more peripheral than the protoconid) and rudimentary distal cuspids; the mesial fovea displays a secondary cristid and is moderately large, although smaller and shallower than the talonid basin; the buccal cingulid is poorly developed. The three lower molar germs (Figs. 3h, i and 4k–m) indicate that the m1 and m2 display a subrectangular and elongated contour and five buccolingually compressed cuspids connected by sharp cristids. The protoconid is more mesial than the metaconid, whereas the hypoconid and entoconid are less peripheral than the mesial cuspids, and the hypoconulid is smaller than the other cuspids and centrally located close to the distal marginal ridge. The mesial fovea is very extensive but slightly wider than long, and displays a secondary transverse cristid that extends from the preprotocristid to the center of the fovea (IPS43936) or to the premetacristid (IPS94886), thus completely dividing the fovea in the latter specimen. The hypoprotocristid emerges distally from the protoconid and angles before meeting the hypometacristid, completely separating the mesial fovea from the more extensive and deeper talonid basin.

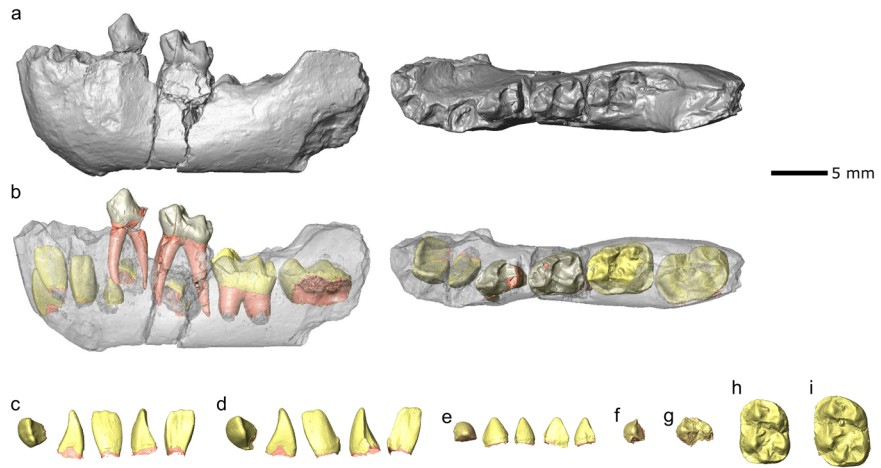

**Fig. 3 | Virtual rendering of the *Pliobates cataloniae* female infant mandible (IPS43936) from ACM/C5-D1 with extracted permanent tooth germs.** Mandible in lateral and occlusal view (**a**), with the bone in semitransparency (**b**) showing the deciduous (light gray) and permanent (yellow) teeth; **c** left i1 in occlusal, mesial, labial, distal, and lingual views; **d** left i2 in occlusal, mesial, labial, distal, and lingual views; **e** left c1 in occlusal, mesial, labial, distal, and lingual views; **f** left p3 in occlusal view; **g** left p4 in occlusal view; **h** left m1 in occlusal view; **i** left m2 in occlusal view. Completed or forming roots are showed in pink.

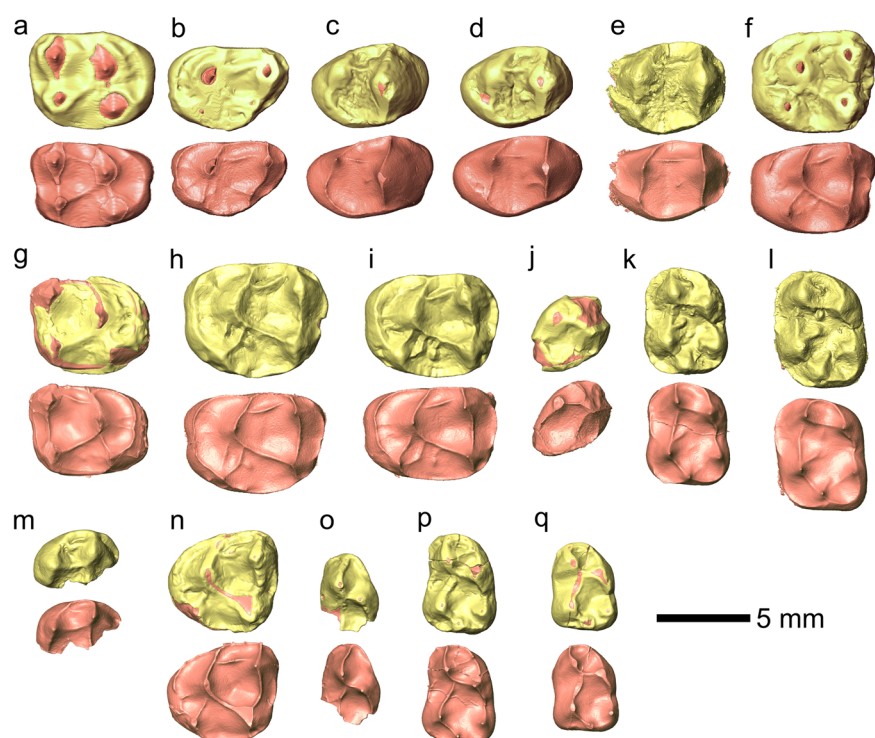

**Fig. 4 | Three-dimensional virtual renderings of the OES and EDJ of *Pliobates cataloniae* cheek teeth from ACM/C5-D1 in occlusal view. a** right M2 (IPS58443.1; holotype); **b** left M3 (IPS58443.2; holotype); **c** left P3 (IPS44014); **d** left P4 (IPS44014); **e** left P4 incompletely formed germ (IPS42977); **f** left M1 (IPS44014); **g** right M1 (IPS100379); **h** left M2 germ (IPS100384); **i** left M2 germ (IPS94888); **j** partial right p4 (IPS93524); **k** left m1 (IPS43936); **l** left m2 (IPS43936); **m** right m2 germ mesial fragment (IPS94886); **n** left dP4 (IPS42977); **o** partial left dp3 (IPS43936); **p** left dp4 (IPS43936); **q** left dp4 (IPS106878). Dentine horns were reconstructed by C.Z. for **a**, **b**, **f**, and **g**. OES and EDJ models are shown in yellow and rose, respectively. Mesial is on top.

The cristid obliqua, constituted by a short postprotocristid and a much longer prehypocristid, is slightly obliquely oriented and directed toward the distal aspect of the protoconid. A well-defined distal arm of the pliopithecine triangle runs mesiolingually into the talonid basin, originating from midway along the prehypocristid (m1) or the mesial aspect of the hypoconid (m2). There is no mesial arm of the plio-pithecine triangle originating from the protoconid, but a short and mesiodistally aligned secondary cristid runs from the hypometacristid (m1) or the center of the transverse cristid (m2) toward the center of the trigonid basin. The distal fovea is located on the distolingual corner of the crown, continuous with the talonid basin (m1) or only partially separated from it (m2). The buccal cingulid is broad and continuous.

The DI1 crown (Fig. 2c′) is somewhat mesially tilted, spatulate, and waisted at the cervix, and displays a moderately ledge-like lingual cingulum. The DP3 (Fig. 2v) has a trapezoidal occlusal contour slightly broader than long and displays three main cusps, with the extensive paracone being transversely aligned with the slightly smaller proto-cone, and the hypocone being much smaller, while there is no distinct metacone. The mesial fovea is restricted whereas the distal basin is very extensive and lacks a crista obliqua; the prehypocrista is directed

to the protocone. The cingula are not well developed. The DP4 (Figs. 2w and 4n) resembles in occlusal morphology the M1–M2 (including the presence of four main cusps, the shape and size of the trigon basin and distal fovea, the presence of a continuous crista obliqua and a C-shaped lingual cingulum, and the peripheral position of the hypocone) but differs in several other respects (more triangular in occlusal contour, more restricted mesial fovea, more rudimentary hypocone, and lack of secondary crest between hypocone and crista obliqua). The dp3 (Figs. 2x and 4o) is mesially very tapering, with a very mesially located protoconid and without a distinct premetacristid, so that the mesial fovea is lingually open. The cristid obliqua, composed of the prehypocristid, is very obliquely inclined. The dp4 (Figs. 2x, y and 4p, q) is more similar to the m1 but displays multiple differences, including a more mesially tapering contour and a longer mesial fovea (albeit similarly divided by a secondary cristid), a more inclined cristid obliqua (directed toward the junction of the hypoprotocristid and hypometacristid; Y-shaped configuration), lack of pliopithecine triangle, and less developed buccal cingulid.

## Morphological comparisons and body mass

The occlusal morphology of *Pliobates* most closely resembles that of crouzeliine pliopithecoids, particularly the indeterminate species from Mörgen as well as *Plesiopliopithecus* and *Crouzelia*. The I1 resembles that of *Barberapithecus*, although the crown is even more asymmetrical in *Pliobates* (unknown in other crouzeliines). The P4 is comparable in shape to that from Mörgen, while the upper molars resemble those of *Barberapithecus* and the Mörgen crouzeliine in the degree of cusp compression and crest sharpness, together with other occlusal details (more distal location of the protocone, i.e., more so than in other taxa, relatively small metacone, and M1 and M2 occlusal contour, small hypocone more peripheral than the protocone, narrow trigon basin, short distal fovea, and lingual cingulum not surrounding the hypocone). Nevertheless, the M1 of *Pliobates* more closely resembles that from Mörgen in the moderate buccolingual wasting and narrow buccal cingulum, resulting in more peripheral buccal cusps. In contrast, the lingual cingulum of *Pliobates* M1 and M2 is broader than in *Barberapithecus* and the Mörgen crouzeliine (unknown in *Plesiopliopithecus* and *Crouzelia*). The M3 hypocone is mesiodistally aligned with the protocone, as in *Barberapithecus* (unknown in other crouzeliines), but the crown is much more trapezoidal and displays a smaller hypocone, narrower cingula, more peripheral buccal cusps, and a more lingually located metacone. As for the lower dentition, the i2 is comparable to that of *Plesiopliopithecus*, while the female c1 is very similar to that of *Barberapithecus*, although relatively broader and with a more marked mesiolingual cuspulid-like enamel thickening. The p4, in contrast, differs from those of other crouzeliines, although the occlusal contour and non-peripheral protoconid approximate the condition of *Plesiopliopithecus* and *Crouzelia*, while the buccal cingulid is as developed as in *Barberapithecus*. The lower molars display the typical crouzeliine pattern, characterized by quite compressed cusps, large foveae, a mesially located protocone relative to the metacone, and a continuous lingual cingulid, except that the mesial arm of the pliopithecine triangle is missing in both the m1 and m2 of *Pliobates*. Among crouzeliines, the lower molars of *Pliobates* more closely resemble *Barberapithecus* (and differ from both *Plesiopliopithecus* and *Crouzelia*) in the well-developed hypoconulid and distal arm of the pliopithecine triangle, and the alignment of the entoconid with the hypoconid. In contrast, the occlusal contour and the distal origin of the hypoprotocristid from the protoconid most closely resemble the condition in *Plesiopliopithecus* and *Crouzelia*, while the distinct hypoentocristid and postcristid can also be found in *Barberapithecus* and *Plesiopliopithecus*. The buccal cingulid is very broad as in *Plesiopliopithecus*, and the distal fovea is not isolated from the talonid basin as in *Crouzelia*. In summary, *Pliobates* closely resembles other

crouzeliines in dental morphology but displays a unique combination of features that supports its distinction at the genus rank.

The dental dimensions of *P. cataloniae* are compared with those of dendropithecids and pliopithecoids in Supplementary Figs. 4 and 5 (see Supplementary Table 2 for more information on the comparative taxa). The upper premolars of *Pliobates* most closely resemble in size and proportions those of *Simiolus* and the crouzeliine from Mörgen, whereas the upper molars approach the condition of *Barberapithecus*, *Dionysopithecus*, and the crouzeliine from Mörgen. In turn, the lower molars of *Pliobates* most closely resemble those of *Crouzelia*, *Barberapithecus*, and *Plesiopliopithecus*. Dental body mass (BM) estimates for *P. cataloniae* range from 4.4 to 5.8 kg, with an average estimated BM of 5.3 kg and an uncertainty degree (based on the combined 50% confidence intervals for each tooth locus) of 3.7 to 7.3 kg (Supplementary Table 3).

## Cladistic analysis

Our phylogenetic analysis based on dental features (Supplementary Data 1 and 2) recovers 192 most parsimonious trees with equal length (tree length = 302). The strict consensus (Fig. 5) recovers pliopithecoids (including *Pliobates*) as a clade of stem catarrhines more basal than dendropithecids but less so than *Saadanius*. Our analysis further recovers the monophyly of dionysopithecids and crouzeliids, but not of pliopithecids, as *Pliopithecus antiquus* appears more closely related to crouzeliids than the remaining pliopithecids, whose relationships are not resolved. The analysis recovers two crouzeliid clades, here distinguished as subfamilies Crouzeliinae and Anapithecinae (Table 2). *Pliobates* is recovered as a crouzeliine, less basal than *Barberapithecus* but more so than *Plesiopliopithecus* and *Crouzelia*, which are recovered as sister taxa. In turn, *Fanchangia* is recovered as the basalmost anapithecine; *Krishnapithecus* and *Laccopithecus* are also recovered as anapithecines, constituting a clade sister to that including *Egarapithecus* and *Anapithecus*.

Based on our cladistic analysis, *Pliobates* possesses the following pliopithecoid synapomorphies (Supplementary Data 3): i2 mesiodistally waisted basally at cervix; female c1 with prominent mesiolingual cristid-lingual cingulid thickening located at about crown-midheight; M1–M2 with paracone not markedly buccally located relative to the metacone; and lower molars (m1–m2) with a distinct distal arm of a pliopithecine triangle. In turn, *Pliobates* displays several synapomorphies of crouzeliids (i2 with clearly inclined mesial margin; P3 with relatively narrow distal basin; p4 with moderately developed buccal cingulid; and very narrow m2) and crouzeliines (P3 longer than P4; molars with buccolingually compressed cusps; M1–M2 with relatively small hypocone; m1–m2 with a large, almost as long as broad, mesial fovea; and entoconid almost transversely aligned with hypoconid (even though this synapomorphy is ambiguous because it would have been reversed in *Plesiopliopithecus* and *Crouzelia*, which display an entoconid clearly more distal than the hypoconid). Finally, among crouzeliids, *Pliobates* most closely resembles *Plesiopliopithecus* and *Crouzelia* in the possession of multiple synapomorphies absent from *Barberapithecus* (female c1 relatively broad with a cuspulid-like mesiolingual cristid-lingual cingulid; molars with and markedly sharp crests; very narrow m1; hypoprotocristid originating distally from protoconid with inclined cristid oblique connecting to where the hypoprotocristid originates, at least in m1; and distal fovea not separated from the talonid basin, at least in m2), but differs from them in the lesser developed buccal cingulid in p4 and the m1 and m2 that show a more distinct distal arm of the pliopithecine triangle, a larger hypoconulid, and an entoconid almost transversely aligned with the hypoconid.

While the dental-only cladistic analysis reported above was devised to determine the closer phylogenetic relationships of *Pliobates* with pliopithecoids, a second cladistic analysis was performed to further investigate the relationships between *Pliobates*, other pliopithecoids,

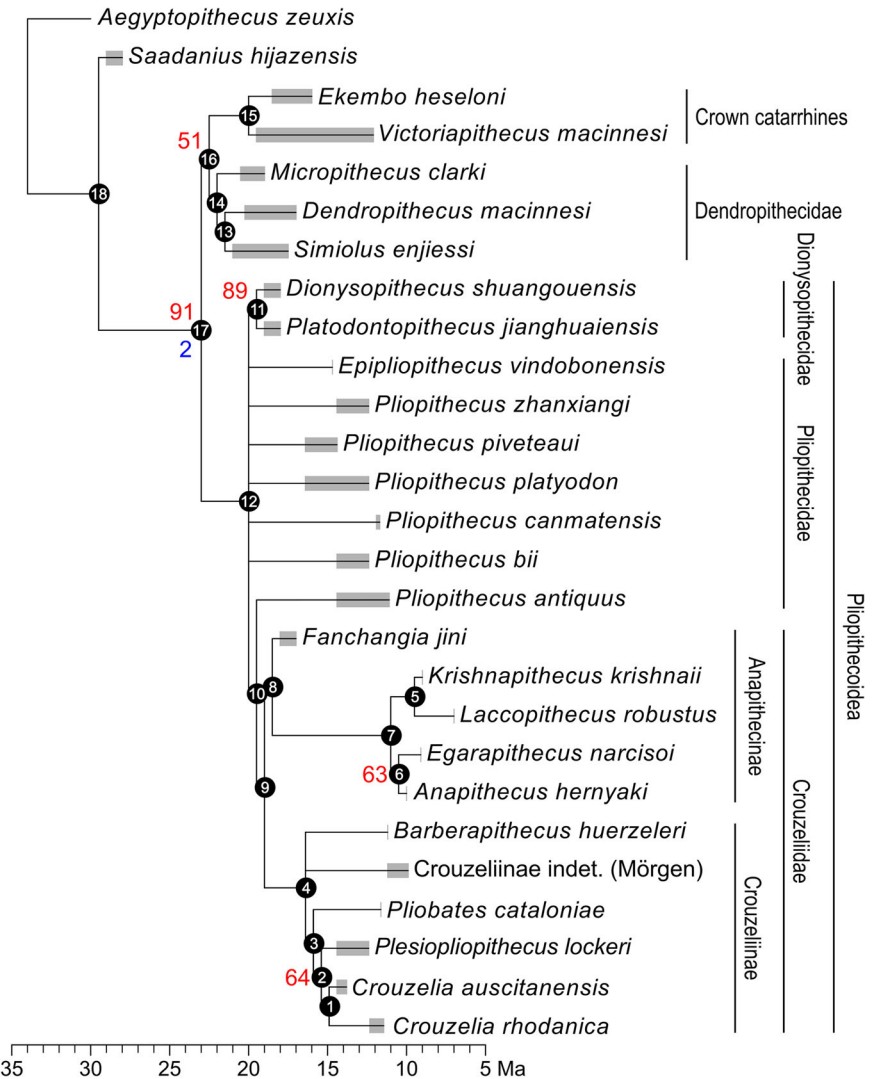

**Fig. 5 | Time-calibrated strict consensus cladogram derived from the 192 most parsimonious trees based on a taxon-character data matrix for 27 taxa and 95 dental characters.** Tree length = 302 steps; CI (excluding uninformative characters) = 0.374; RI = 0.505; RCI = 0.189. Bremer's indices (only shown when >1) and bootstrap percentages (only shown when ≥50%) are reported next to the nodes in blue and red, respectively. Gray semitransparent rectangles represent the chronostratigraphic ranges (including dating uncertainties) of the depicted taxa (Supplementary Table 5). Internal nodes have been depicted arbitrarily 0.5 Myr before the oldest record of the group or the next oldest node included within that clade. The list of characters and taxon-character matrix can be found in Supplementary Data 1 and 2, respectively. Node numbers within black dots refer to clades in the list of apomorphies reported in Supplementary Data 3.

and hominoids. This analysis, run at the genus level, included a more restricted sample of pliopithecoids but a wider array of hominoids and further considered cranial and postcranial characters. The results differ depending on whether postcranial characters are included (Supplementary Text 4). The iterations based on dental and craniodental characters (Supplementary Fig. 6a, b) yield similar results to the more detailed dental-only analysis of pliopithecoids at the species level reported above. In particular, pliopithecoids are recovered as a stem catarrhine clade and the monophyly of crouzeliids is supported, with *Pliobates* being a crouzeliid more derived than *Barberapithecus*. In contrast, the iteration exclusively based on postcranial features only adequately resolves the phylogeny of the crown hominoid clade and recovers *Pliobates* as a stem hominoid more derived than *Ekembo* (Supplementary Fig. 6c). Finally, the iteration based on both craniodental and postcranial evidence yields a relatively well-resolved but very heterodox topology (Supplementary Fig. 6d), in which *Pliobates* still branches among crouzeliids but the position of *Ekembo* remains unresolved and dendropithecids and pliopithecoids are recovered as successive paraphyletic assemblages of stem hominoids.

## Discussion

### The systematic position of *Pliobates* based on dental morphology

*Pliobates* was originally proposed as a stem hominoid[1] but subsequently recovered as a pliopithecoid[2,3] or even as a more basal stem catarrhine[8] by cladistic analyses including craniodental and postcranial features (Fig. 1; Supplementary Fig. 7). Dental morphology—which provides the main basis for the taxonomy of fossil primates and most other groups of mammals, and is highly diagnostic for pliopithecoids—offers the prospect to settle the debate on the systematic position of *Pliobates*. Its dental morphology did not play a central role in the original description because available remains were not particularly informative, merely noting dental similarities with dendropithecids[1], particularly to the upper molars of *Micropithecus*. However, some of these features (e.g., markedly convex lingual profile with a C-shaped lingual cingulum that does not surround the hypocone) are not exclusive to these taxa. In contrast, the newly reported molars (especially the lower ones) show much closer resemblances to crouzeliines. At the same time, *Pliobates* displays several differences relative to

## Table 2 | Updated classification of pliopithecoids to the sub-family rank based on the published literature[8,20], and this study

| |
|---|
| Superfamily Pliopithecoidea |
| Family Dionysopithecidae |
| Genus *Dionysopithecus* |
| *Dionysopithecus shuangouensis* |
| *Dionysopithecus orientalis* |
| Genus *Platodontopithecus* |
| *Platodontopithecus jianghuaiensis* |
| Family Pliopithecidae* |
| Genus *Epipliopithecus* |
| *Epipliopithecus vindobonensis* |
| Genus *Pliopithecus* |
| *Pliopithecus antiquus* |
| *Pliopithecus bii* |
| *Pliopithecus canmatensis* |
| *Pliopithecus piveteaui* |
| *Pliopithecus platyodon* |
| *Pliopithecus zhanxiangi* |
| Family Crouzeliidae |
| Subfamily Anapithecinae |
| Genus *Anapithecus* |
| *Anapithecus hernyaki* |
| ?*Anapithecus priensis* |
| Genus *Egarapithecus* |
| *Egarapithecus narcisoi* |
| Genus *Fanchangia* |
| *Fanchangia jini* |
| Genus *Krishnapithecus* |
| *Krishnapithecus krishnaii* |
| Genus *Laccopithecus* |
| *Laccopithecus robustus* |
| Subfamily Crouzeliinae |
| Genus *Barberapithecus* |
| *Barberapithecus huerzeleri* |
| Genus *Crouzelia* |
| *Crouzelia auscitanensis* |
| *Crouzelia rhodanica* |
| Genus *Plesiopliopithecus* |
| *Plesiopliopithecus lockeri* |
| Genus *Pliobates* |
| *Pliobates cataloniae* |

Taxa denoted with an asterisk are likely paraphyletic. *Pliopithecus antiquus* is provisionally retained within the Pliopithecidae despite our cladistic results supporting a closer link with the Crouzeliidae (rendering the former paraphyletic). Based on our results, *Krishnapithecus* is included within the Anapithecinae (with Krishnapithecidae being a junior synonym of Crouzeliidae). *Kapi* might also belong to the Anapithecinae, given the similarities with *Krishnapithecus* noted by Ji et al.[8], but we prefer to leave it as incertae sedis at the superfamily rank given that it was not included in our cladistic analysis. The Mörgen material is left unassigned to genus given the lack of lower teeth.

previously known crouzeliine genera that warrant a genus distinction, while the generic ascription of the material from Mörgen[23,24] cannot be adequately determined because it cannot be directly compared with *Crouzelia* and *Plesiopliopithecus*.

The cladistic analysis of dental features at the species level reported here, including a wide representation of pliopithecoid taxa, further supports that *Pliobates* is a crouzeliid pliopithecoid most closely related to *Crouzelia* and *Plesiopliopithecus*. The recovery of pliopithecoids

(including *Pliobates*) as a clade of stem catarrhines more basal than dendropithecids is consistent with some[2,3,8] but not all[1,8] previous results. Nevertheless, our results support that *Barberapithecus* is not a pliopithecid[18] but a crouzeliid[1,20,25] more closely related to crouzeliines[20] than to anapithecines[25]. *Krishnapithecus*, previously included in a distinct pliopithecoid family[8,20,26], appears more closely related to anapithecines—Krishnapithecidae thus being here considered a junior subjective synonym of Crouzeliidae. The nesting of *Pliobates* within the crouzeliine clade further implies that Pliobatidae must be also considered a junior subjective synonym of the Crouzeliidae.

A second cladistic analysis of (cranio)dental features at the genus level, including a wider representation of hominoids, further recovers *Pliobates* as a crouzeliid and supports pliopithecoids as a clade of stem catarrhines, but yields a somewhat different topology regarding *Anapithecus*. Postcranial features alone are not informative enough about the phylogeny of Miocene catarrhines (probably due to large amounts of missing data) but in contrast to craniodental evidence support a stem hominoid status for *Pliobates*. When postcranial features are combined with craniodental data, *Pliobates* branches again among crouzeliid genera but dendropithecids and pliopithecoids as a whole are recovered as successive paraphyletic assemblages of stem hominoids instead of stem catarrhines, strongly resembling the cladistic results accompanying the original description of *Pliobates*[1]. Given the paraphyletic status of pliopithecids in our dental-only analysis at the species level and the different topologies of *Anapithecus* in the first and second cladistic analyses, further research would be required to clarify the internal phylogeny of pliopithecoids and their relationships with other extinct and extant catarrhines. The contrasting results obtained by (cranio) dental vs. craniodental + postcranial evidence reported here parallel the contradictory results obtained by previous cladistic analyses including *Pliobates*[1-3,8]. In our opinion, these contradictions stem from the fact that *Pliobates* displays a mosaic of primitive (stem catarrhine-like) and derived (crown hominoid-like) features[1], particularly in the postcranium, so that different topologies may be favored depending on the emphasis put on these features. Although the results based on all available evidence should be preferred on cladistic epistemological grounds, the fact that postcranial features alone yield a completely different topology for *Pliobates* and that postcranial homoplasy has repeatedly been noted as misleading in hominoid phylogenetics[11-15] suggest otherwise. The cranial morphology of *Pliobates* is also ambiguous because some similarities with hylobatids (anteriorly situated orbits, broad interorbital distance, short face, low zygomatic roots) are also displayed by the pliopithecoid *Epipliopithecus*[27], stem hominoids such as nyanzapithecids[2], and small-bodied catarrhines from East Africa such as the dendropithecid *Micropithecus*[28]. It is thus likely that the hylobatid-like features of *Pliobates* are either symplesiomorphic for crown catarrhines and/or homoplastic among pliopithecoids, hylobatids, and some stem hominoids, maybe being related to small body size[29]. However, while the inclusion of cranial characters does not alter the topology recovered on the basis of teeth, the postcranial dataset has a major influence on the most parsimonious topology, suggesting that, due to rampant homoplasy, it is introducing more 'noise' than phylogenetic signal in the case of *Pliobates*.

### Implications for pliopithecoid locomotor diversity

Part of the problem when it comes to include postcranial characters in phylogenetic analyses of pliopithecoids is undoubtedly attributable to the large amount of missing data for most species. Besides *Pliobates*, which is now the most complete crouzeliid individual known, it is only well known in the pliopithecid *Epipliopithecus*[23,30,31], generally interpreted as a generalized quadruped with limited climbing and suspensory abilities[32-34]. However, the meager available record suggests that the group as a whole displayed much more diverse postcranial adaptations[34-36]. This is most clearly evidenced by the postcranium of *Pliobates*, which reveals a mosaic of stem catarrhine-like features

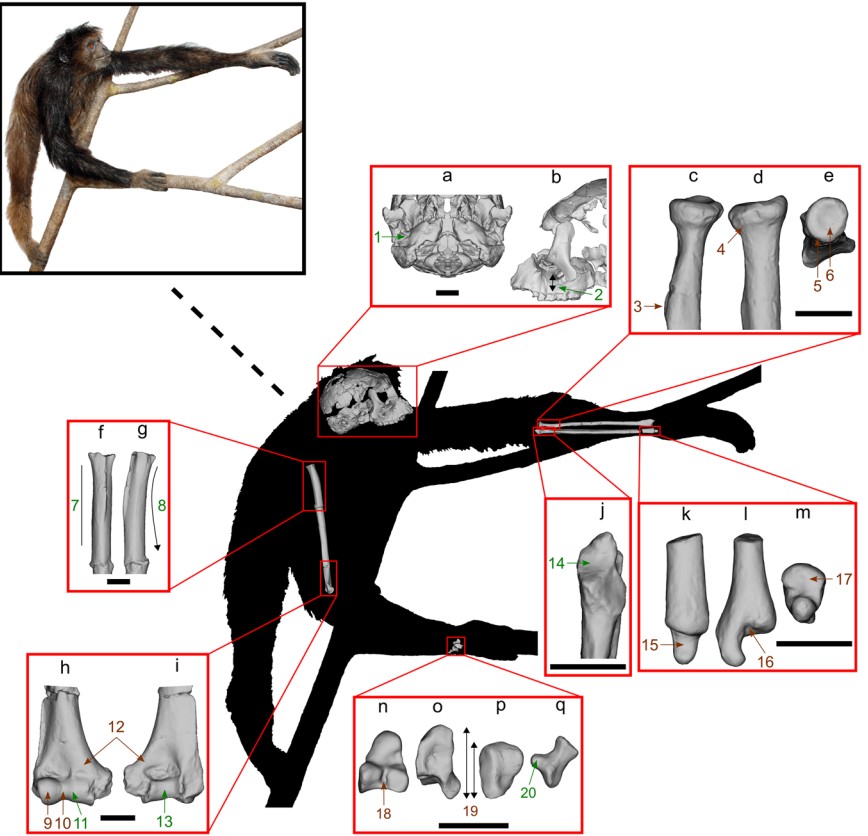

**Fig. 6 | Life reconstruction of *Pliobates* in color and as a black silhouette indicating key morphological features that highlight its mosaic of primitive and derived features relative to crown hominoids. a**, **b** cranium in inferior (**a**) and lateral (**b**) views; **c**–**e** left proximal radius in anterior (**c**), medial (**d**), and proximal (**e**) views; **f**–**g** left proximal humerus (mirrored) in anterior (**f**) and medial (**g**) views; **h**–**i** left distal humerus (mirrored) in anterior (**h**) and posterior (**i**) views; **j** left proximal ulna in anterior view; **k**–**m** left distal ulna in anterior (**k**), medial (**l**), and distal (**m**) views; **n** left capitate (mirrored) and **o** left hamate (mirrored) in radial views; **p** left triquetrum (mirrored) in proximal view; **q** right pisiform in radial view. Green numbers denote plesiomorphic (stem catarrhine-like) characters whereas brown numbers denote derived (crown hominoid-like) features, and are as follows: 1, short

and incompletely ossified tubular ectotympanic; 2, low zygomatic root; 3, laterally facing bicipital tuberosity; 4, slightly tilted head with reduced lateral lip; 5, beveled surface for the zona conoidea; 6, almost circular head; 7, straight shaft; 8, somewhat retroflexed shaft proximally; 9, moderately globular capitulum; 10, well-developed zona conoidea; 11, poorly defined trochlear lateral keel; 12, lack of entepicondylar foramen; 13, lack of spool-shaped trochlea; 14, narrow trochlear notch without a median keel; 15, slender and hook-like styloid process; 16, more developed ulnar fovea; 17, expanded head with a two-faceted semilunar articulation; 18, facet for the second metacarpal divided by a deep groove for the carpometacarpal ligament; 19, small triquetrum relative to hamate size; 20, small facet for the styloid process. Scale = 1 cm.

combined with crown catarrhine and even crown hominoid synapomorphies absent in *Epipliopithecus*[1] (see Fig. 6 and Supplementary Table 4), such as modern hominoid-like humeroradial and diarthrodial distal radioulnar joints[1]. Previous inferences about the positional behavior of crouzeliid pliopithecoids were very restricted, based on a few isolated specimens available for each genus, suggesting (semi) arboreal quadrupedalism with strong climbing abilities and a varying degree of suspensory behaviors[18,34,35]. Based on the phalanges and femora of *Anapithecus*[18,37] and especially a phalanx of *Laccopithecus*[38], anapithecines may be inferred as more suspensory than *Epipliopithecus*. In turn, a talus and a calcaneus of *Crouzelia* indicate a strong quadrupedal component[35], whereas a proximal radius of the stem crouzeliine *Barberapithecus* indicates enhanced mobility at the elbow joint compared with *Epipliopithecus* but evinces no clear climbing or suspensory adaptations[34]. In contrast, the hominoid-like humeroradial articulation of *Pliobates*, characterized among others by a moderately globular capitulum, a well-developed zona conoidea, and an almost circular radial head with a reduced lateral lip, is clearly adapted for climbing, while its reduced ulnocarpal articulation (more derived than in *Ekembo*) indicates a decreased emphasis on quadrupedalism[1]. Overall, the mosaic postcranial morphology of *Pliobates* is compatible with a versatile arboreal positional repertoire combining significant amounts of eclectic and cautious climbing with

above-branch quadrupedalism and some degree of below-branch suspensory behaviors[1].

## Implications for ape evolution

Several decades ago, pliopithecoids were considered broadly ancestral to hylobatids (Supplementary Fig. 7) because of resemblances in cranial morphology as well as body size and proportions[27,31,39,40]. However, since the spread of cladistics in the 1980s, pliopithecoids have been generally regarded as stem catarrhines more derived than the Early Oligocene propliopithecoids from Afro-Arabia but less so than dendropithecids[19] (Supplementary Fig. 7). Our cladistic analyses support *Pliobates* as a crouzeliid pliopithecoid but yield different results regarding the systematic status of pliopithecoids depending on whether postcranial characters are considered. In that case, *Pliobates* and other pliopithecoids are supported as members of the stem hominoid lineage, as in some[1,4] but not most[2,3,5,41] previous cladistic analyses, which support instead a stem catarrhine status. The latter is supported by the lack of crown catarrhine synapomorphies coupled with the retention of more plesiomorphic features including an incompletely ossified ectotympanic, the presence of entepicondylar foramen in the distal humerus, and the possession of a single hinge-like carpometacarpal thumb joint[16–19,42–46]. *Pliobates* is more derived than pliopithecoids by lacking the entepicondylar foramen and displaying

multiple hominoid-like elbow and wrist specializations[1], but its short tubular and incompletely ossified ectotympanic[1], less derived than hose of the stem cercopithecoid *Victoriapithecus*[47] and the stem hominoids *Ekembo*[48] and *Nyanzapithecus*[2], is more similar to that of *Epipliopithecus*[23] and thus more consistent with a stem catarrhine status.

Deciphering the phylogenetic relationships among pliopithecoids, crown catarrhines, and other Miocene catarrhines is not the main aim of this work but the results of our cladistic analyses support the view that pliopithecoids (including *Pliobates*) are stem catarrhines and that previous results indicating a stem hominoid status[1] are probably attributable to the independent acquisition in crouzeliids of postcranial ape-like features. This interpretation illustrates the confounding effect that postcranial convergences might also have when inferring the internal phylogeny of living and fossil hominoids[11–15], not only between extinct taxa such as *Oreopithecus* and crown hominids, but also between hylobatids and hominids. The spread of cladistics promoted the view that postcranial similarities between these clades are synapomorphic[45,49], but the discovery that the pongine *Sivapithecus* displayed a much more primitive postcranium than expected[50] reopened this debate during the 1990s[11]. Subsequent discoveries of *Sivapithecus*[51] and the stem hominid *Pierolapithecus*[52,53] further reinforced the previous contention[11,54] that such similarities may be largely homoplastic[12–15]. However, cladistic analyses performed during the last decades[14,55,56] have continued to support a close relationship between hylobatids and hominids to the exclusion of all Early and most Middle Miocene apes despite molecular evidence indicating that they diverged during the Early Miocene[57,58]. Hominoid phylogenetic inference might be affected by a problem of long-branch attraction between hylobatids and hominids due to the numerous postcranial similarities presumably evolved in parallel between these lineages[15]. Uncertainties surrounding the origin of hylobatids and the lack of associated postcrania for many Miocene apes precludes adequately testing this hypothesis, which is nevertheless supported by preliminary results of tip-dating Bayesian analyses (Fig. 4.3 in ref. [59]) and the contradictory results provided by craniodental and postcranial evidence regarding the branching of hylobatids relative to many Miocene apes (Fig. 4 in ref. [14]).

The existence of evolutionary convergences in multiple postcranial features among crown hominoids, *Pliobates*, and more distantly related primates such as atelids does not necessarily imply that these features cannot be truly synapomorphic between hylobatids and hominids. However, our cladistic analyses and the more plesiomorphic condition displayed by *Epipliopithecus* support that the crown hominoid-like features of the elbow and wrist joints in *Pliobates* were independently acquired. Given some similarities with *Ekembo*, which lacks suspensory adaptations but displays a rather modern humeroradial joint[11] and an incipient distal radioulnar diarthrosis[60], the condition of *Pliobates* for these joints may still represent a good analog (not homolog) for an ancestral morphotype from which crown hominoids evolved, supporting the view that the aforementioned features originally evolved as climbing adaptations that were later co-opted for suspensory behaviors[61,62] in crown hominoids. The presence of purported crown hominoid synapomorphies—functionally related to orthograde behaviors (i.e., those performed with the trunk held vertically, such as vertical climbing and suspension)—in a pliopithecoid stem catarrhine should not be surprising because climbing adaptations similar to those of apes are also present in lorisids[61,62], while suspensory adaptations have repeatedly evolved in distantly related taxa other than hominoids, such as atelid New World monkeys[11,63,64] and sloth lemurs[65]. Nevertheless, the possession of such features in *Pliobates* renders additional plausibility to the hypothesis that they could have also independently evolved between hylobatids and hominids, as argued in multiple studies[11–15], and as traditionally assumed before the advent of cladistics[11,48,66]. In particular, *Pliobates*

illustrates how easily the inclusion of postcranial features prone to homoplasy may override phylogenetic signal—in the case of *Pliobates*, recovering all pliopithecoids as stem hominoids instead of stem catarrhines, as supported instead by (cranio)dental features. In summary, the pliopithecoid status of *Pliobates* supports that similar postcranial features have repeatedly evolved among various catarrhine lineages—likely owing to similar locomotor selection pressures related to orthograde behaviors—and highlights the dramatically misleading (and often underrated) effects that such convergences might have on most parsimonious cladograms in hominoid phylogenetics.

## Methods
No relevant ethical regulations were required for the present study.

### Age and geological background
The newly described remains come from locality ACM/C5-D1, which is located on meter 188 of the 300-m-thick composite stratigraphic sequence of ACM (els Hostalets de Pierola, Vallès-Penedès Basin, NE Iberian Peninsula), being correlated to chron C5r.2n with an interpolated age of 11.63 Ma[21,22], which roughly coincides with the Middle/Late Miocene boundary. This locality is thus only minimally older than the type locality of *P. cataloniae* (ACM/C8-A4), which is located stratigraphically 6 m above ACM/C5-D1 within the same chron, with an interpolated age of 11.62 Ma[21,22]. Both localities were located close to each other (Supplementary Figs. 8 and 9). On biostratigraphic grounds, they are correlated to the *Democricetodon crusafonti* – *Hippotherium* interval subzone of the Vallès-Penedès Basin[67], MN7 + 8, late Aragonian[22].

### MicroCT scan acquisition and segmentation
IPS58443 was originally scanned at a resolution of 95 μm[1]. To explore dental structures, the specimen was scanned again by X-ray microtomography at the Centro Nacional de Investigación sobre la Evolución Humana (CENIEH; Burgos, Spain) using a GE Phoenix V|Tome|X s240 mCT scanner with the following parameters: 0.35 mA current, 170 kV voltage, 0.2 mm Cu filter, and a magnification of 9.52. The final reconstructed volume has an isometric voxel size of 33 μm. Nine specimens from ACM/C5-D1 were also scanned by X-ray microtomography at the CENIEH with the following parameters: 120 mA current and 115 kV voltage, 0.2 mm Cu filter, and a magnification of 16.67. The final reconstructed volumes have an isometric voxel size of 12 μm. The μCT scans were segmented using Avizo v.7.0 (Visualization Sciences Group, Mérignac) to digitally reconstruct the EDJ surface.

### Dental measurements, dental nomenclature, and comparative sample
Dental size measurements of *P. cataloniae* consist of mesiodistal length (MD) and buccolingual breadth (BL), the latter measured separately at the mesial (BLm) and distal (BLd) lobes in the case of molars and dp4. These measurements were taken to the nearest 0.1 mm with a digital caliper in all specimens except for the P4 of IPS42977 and the m1 and m2 of IPS43936, which had to be digitally extracted from microCT scans and were thus measured using the 2D Length tool in Avizo v. 7.0. The breadth/length index (BLI, in %) was computed for cheek teeth as BLI = maximum BL/MD × 100 in Microsoft Excel 2023. Dental nomenclature follows Harrison and Gu[17]. Dental measurements for the comparative sample (including dendropithecids and pliopithecoids) were taken from published sources and/or measured from casts by D.M.A. (see Supplementary Table 2). Details on all taxa analyzed are provided in Supplementary Table 5.

### Systematics
A modified classification is proposed here for pliopithecoids, which updates the previously available classification from Harrison et al.[20] based on recent results for *Kapi*[8] and the results of our cladistic analysis (Fig. 5; Table 2).

## Body mass estimation

The body mass (BM, in kg) of *P. cataloniae* was estimated from dental measurements of upper and lower molars (M1–M3 and m1–m2) using anthropoid allometric equations based on occlusal tooth square area (A, in mm²)[68], computed as $A = MD \times BL$. For tooth positions represented by several specimens, average A was used to estimate BM. Logarithmic detransformation bias was corrected using the ratio estimator[68]. The 50% confidence interval (CI) was computed for each tooth locus using the standard error of estimate and an inverse Student's *t* distribution with the degrees of freedom determined by the effective sample size[68]. An average estimate was computed as the arithmetic mean of estimates for the various tooth loci, with the CI taken as the maximum and minimum ends of the 50% CIs of the included tooth loci. All computations were performed in Microsoft Excel 2023.

## Cladistic analysis

The phylogenetic relationships of *P. cataloniae* were inferred by means of a cladistic analysis based on maximum parsimony using PAUP* v. 4.0a169[69]. A character-taxon matrix including all pliopithecoids, three dendropithecid species, and two crown catarrhine species, along with the stem catarrhines *Aegyptopithecus zeuxis* (used as outgroup) and *Saadanius hijanzensis*, was compiled, using and/or modifying dental characters from published sources[1,8,20] and defining new ones (see Supplementary Data 1 for character statements). All characters were treated as unordered, variable characters were scored as multistate, and continuous characters were discretized using the gap-weighted coding method[70]. 23 characters were continuous and discretized. 9 characters were parsimony uninformative. A heuristic search method was applied to search for most parsimonious trees and a strict consensus tree computed. The following metrics were computed: consistency index (CI) excluding uninformative characters, retention index (RI), and rescaled consistency index (RCI). Clade robusticity was assessed by means of bootstrap analysis (1000 replicates) and Bremer's support indices. The analyzed pliopithecoid taxa are the following (Supplementary Table 5): *Dionysopithecus shuangouensis* and *Platodontopithecus jianghuaiensis* from China; *Fanchangia jini* from China; *Pliopithecus* spp. from Europe and China; *Epipliopithecus vindobonensis* from Slovakia; *Crouzelia auscitanensis* and *Plesiopliopithecus lockeri* from France and Austria, respectively; *Crouzelia rhodanica* from France; *Barberapithecus huerzeleri* from Spain; *Anapithecus hernyaki* from Austria, Germany, and Hungary; the Mörgen crouzeliine from Germany; *Egarapithecus narcisoi* from Spain; *Krishnapithecus krishnaii* from India; and *Laccopithecus robustus* from China. Other analyzed taxa include the stem cercopithecoid *Victoriapithecus macinnesi* from Kenya and Uganda, the stem hominoid *Ekembo heseloni* from Kenya, the stem catarrhine *Saadanius hijazensis* from Saudi Arabia, and the dendropithecids *Dendropithecus macinnesi* from Kenya, *Micropithecus clarki* from Uganda, and *Simiolus enjiessi* from Kenya.

A second cladistic analysis at the genus level, essentially based on a modified version of the dental features used for the dental analysis at the species level but further including cranial and postcranial characters, and with a more restricted representation of pliopithecoids but a wider sample of both extant and extinct hominoids, was performed. The taxa included are those used in the analysis in ref. 1 with some improvements: *Proconsul* s.l. was replaced by *Ekembo* (i.e., excluding *Proconsul* s.s.), the dendropithecids *Simiolus* and *Dendropithecus* were kept separate, and the the crouzeliines *Crouzelia* and *Plesiopliopithecus* were analyzed separately as well. We further grouped all *Pliopithecus* species (as in ref. 1), implying that some dental characters were coded as multistate. Regarding the dental features, we added two additional characters, #96 and #97, corresponding to characters #4 and #40 in ref. 1 (Supplementary Data 4 and 5); furthermore, due to the addition of new taxa, we modified the definition and/or recoded the taxa for the

following characters: #18, #24, #35, #38, #41, #43, #45, #47, #53–56, #58–59, #64–69, and #88 (Supplementary Data 4 and 5). We further updated the cranial and postcranial features based on recently published papers dealing with *Pliobates* and/or other pliopithecoids[9,10,34,46] (Supplementary Data 4 and 5): in the case of the cranium, we updated the characters related to the carotid canal based on ref. 9 (i.e., discarding characters #106 and #107 of ref. 1 and rather adding the two characters from ref. 9 corresponding to #167 and #168 here; Supplementary Data 4 and 5) and added those of the semicircular canals based on ref. 10 and the preliminary results reported so far for *Pliobates* in ref. 46 (characters #169–175; Supplementary Data 4 and 5). For the postcranial characters, we only modified some of the character states of *Barberapithecus* proximal radius following the results of ref. 34 (i.e., #143–145, #150–151, and #156–157 in ref. 1, corresponding to #205–207, #212–213, and #218–219 here; Supplementary Data 4 and 5). The rationale behind the use of the cranial and postcranial characters and taxa analyzed in ref. 1 as a base for this second cladistic analysis is that the analysis of ref. 1 is the only that thus far recovered *Pliobates* as a stem hominoid instead of a pliopithecoid. Similar parameters to those of the dental analysis were applied (i.e., character types and scoring, search methods, and metric computation). In total, 22 characters were parsimony uninformative. Given the suspicion that postcranial homoplasy might be distorting the most parsimonious topology recovered for hominoids by parsimony analyses[14,15], we decided to analyze dental, craniodental, and postcranial characters both jointly and separately, to evaluate the influence of each of these anatomical regions on the results. In the analysis based exclusively on postcranial characters, *Saadanius*, *Dionysopithecus*, *Plesiopliopithecus*, and *Crouzelia* were removed because of the excessive amount of missing data for the characters considered in ref. 1. Some dental measurements necessary to code some features in extant taxa were downloaded from PRIMO, the NYCEP PRImate Morphology Online database (http://primo.nycep.org).

## Reporting summary

Further information on research design is available in the Nature Portfolio Reporting Summary linked to this article.

## Data availability

All dentognathic remains of *P. cataloniae*, including those from ACM/C5-D1 and the holotype, are housed in the Institut Català de Paleontologia Miquel Crusafont (ICP), Sabadell, Spain, and available to study by other researchers. 3D models of the outer enamel surface (OES) are openly available from MorphoSource (Supplementary Table 6), whereas 3D models of the enamel-dentine junction (EDJ) have been uploaded to MorphoSource (Supplementary Table 6) but are embargoed until ongoing research of tooth endostructural morphology is published. MicroCT scans are curated at the ICP and have also been deposited in MorphoSource. They are accessible upon reasonable request by other researchers, as in the case of the physical fossils housed in the same institution. Most dental measurements for the comparative sample (including both extinct and extant taxa) are available from various published sources. Some dental measurements for extant taxa are available from PRIMO, the NYCEP PRImate Morphology Online database (http://primo.nycep.org). The authors declare that all other data supporting the findings of this study are available within the paper, its supplementary information files, and supplementary data files. Source data are provided as a Source Data file. Source data are provided with this paper.

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

## Acknowledgements

This publication is part of R + D + I projects PID2020-116908GB-I00 (to S.M.S. and J.M.R.), PID2020-117289GB-I00 (to D.M.A. and J.M.R.), and PID2020-117118GB-I00 (to J.F.), funded by the Agencia Estatal de Investigación of the Ministerio de Ciencia e Innovación from Spain (MCIN/AEI/10.13039/501100011033/). Research has also been funded by the Generalitat de Catalunya/CERCA Programme (to F.B., A.U., S.A., J.F., J.M.R., S.M.S. and D.M.A.); the Agència de Gestió d'Ajuts Universitaris i de Recerca of the Generalitat de Catalunya (Consolidated Research Groups 2022 SGR 00620 to D.M.A. and J.M.R., 2022 SGR 01184 to J.F., and 2022 SGR 01188 to S.M.S.); the Departament de Cultura of the Generalitat de Catalunya (CLT009/18/00071 to S.M.S. and CLT0009_22_000018 to D.M.A.); a predoctoral grant from the Ministerio de Ciencia e Innovación (PRE2018-083299 to F.B.); a Margarita Salas postdoctoral fellowship funded by the European Union NextGenerationEU to A.U.; and a Ramón y Cajal grant (RYC2021-032857-I) financed by the Agencia Estatal de Investigación of the Ministerio de Ciencia e Innovación from Spain (MCIN/AEI/10.13039/501100011033) and the European Union «NextGenerationEU» / PRTR to J.F. Fieldwork was defrayed by CESPA Gestión de Residuos and UTE Ampliació Can Mata. We further acknowledge the support of the Servei d'Arqueologia i Paleontologia of the Generalitat de Catalunya and the collaboration of the Centre d'Interpretació i Restauració Paleontològica (Ajuntaments dels Hostalets de Pierola). μCT scans of *Pliobates* were performed at the CENIEH facilities (Burgos, Spain) with the collaboration of the CENIEH staff. We thank M. Méndez and other ICP technicians for their efforts in screen-washing and concentrate sorting of sediments from ACM/C5-D1, Sergio Llácer for the segmentation of most of the dental specimens of *Pliobates*, E. Delson for granting us access to the PRIMO database and his collection of casts at the AMNH, J. Galindo for collection assistance, S.G. Arranz for assistance with photographs, the staff of the Preparation Area of the ICP for the excellent preparation of the described specimens, and all the field paleontologists that worked at ACM throughout the years for their endurance.

## Author contributions

F.B., S.M.S. and D.M.A. designed research and assembled the cladistic data matrix; F.B., performed the phylogenetic analyses; F.B. and D.M.A. performed the descriptions and comparisons; J.F. supervised microCT-scanning and segmentation; C.Z. contributed to the segmentations and reconstructions of some of the virtual models; J.M.R. directed fieldwork and performed stratigraphic correlations; C.Z., A.U., S.A., J.F., A.B., S.M.S. and D.M.A. contributed to the interpretation of the results; S.A. contributed to the design of some of the figures; F.B., S.A, and D.M.A. wrote the paper with input from all other authors.

## Competing interests

The authors declare no competing interest.
