## [Peer Review File · Nature Communications]

The Miocene primate Pliobates is a pliopithecoidEDITORIAL NOTE: Some text has been redacted to remove confidential information that did not influence the revisions or editorial decisions.

REVIEWER COMMENTS

Reviewer #1 (Remarks to the Author):

This is an excellent study that presents an important reanalysis of the phylogenetic relationships of Pliobates from the Late Miocene of Spain. Pliobates, one of the best known European Miocene catarrhines, has previously been reported to be a stem hominoid, but the current work provides a re-evaluation of the evidence, including new fossil material, that more convincingly shows it to be a crouzeliid pliopithecoid. The study represents a significant advance in our understanding of Miocene catarrhines in Eurasia. The description of the morphology is detailed and accurate, and the phylogenetic analysis is sound and includes all of the pertinent taxa. I do not have any major critical comments to offer in terms of the reanalysis of the phylogenetic relationships or the general conclusions reached. I recommend publication subject to minor revisions.

There are a few issues to take into consideration before the manuscript is finalized for publication.

(1) The authors contend, at several points in the text, that their finding “support the independent acquisition of postcranial similarities between lesser and great apes”. However, this is not a logical assumption or interpretation, and the claims made here are overstated. Finding parallelisms in an outgroup does not have a direct bearing on the likelihood of homoplasy in the hominoid crown group. One could argue that the independent acquisition of similar morphological features associated with forelimb suspension in multiple unrelated lineages (such as sloth lemurs, atelid primates, and sloths) implies that this could be one plausible evolutionary pathway by which hominoid acquired their specialized postcranial morphology, but it does not represent a valid test of that outcome. In this case, Pliobates does not offer evidence that is directly relevant to the question of hominoid postcranial homoplasy - it merely provides a further example of parallel adaptation in a primate as a consequence of similar functional demands.

(2) The text and tables refer to the crouzeliid from Mörge. This is an interesting specimen, but the authors initially identify it as an “indeterminate species”. However, in the Figures and Tables it is identified as cf. *Pliobates*. This seems to be a taxonomic assessment that requires a more detailed comparative study and further justification. It is particularly problematic because the phylogenetic analysis does not link it as the sister taxon of *Pliobates cataloniae*. It would be best to delete reference to this specimen, since it adds nothing of significance to the overall findings and may be confusing to the general reader.

(3) I would recommend that the authors reassess the way in which they analyzed and presented the estimated body mass of *Pliobates*. The authors use the dental metric/body mass allometric equations published by Egi et al. (2004). These regressions and equations are based on species average data, so they provide a reasonable estimation of the average body mass of extinct species. However, applying the equations to estimate the body mass of individual specimens and to use these data to generate intraspecific ranges of body mass or degree of sexual dimorphism may not be valid.

(4) The authors state that the “presence of purported crown hominoid synapomorphies” are “functionally related to orthograde behaviors”. I would contend that the synapomorphies relate to orthograde and forelimb suspension, not just orthograde (because the two positional behaviors are not always functionally linked).

A few minor corrections and clarifications:

139 & Fig. 3: “Infantile” should read “infant”

357 & Fig. 6: “Globulous” should read “globular”

Table 1: The legend needs to explain abbreviation of “BLI”

736: “Unfinished germ”?

Fig. 5: Spelling of *Platodontopithecus* and *krishnaii*

759: “Low zygomatic root” is ambiguous. Better described, perhaps, as “low zygomatic arch root”?

765: Spelling “groove”

REDACTED

Reviewer #2 (Remarks to the Author):

Comments

Pliobates is a small-bodied catarrhine primate discovered from the middle/late Miocene boundary of Spain. In the first description of this genus, it was suggested to be a stem hominoid before the divergence between the hylobatids and hominids, being more derived than the East African stem hominoids (proconsulids). If this is the case, Pliobates gives a great influence on our understanding of the last common ancestor of gibbons and great apes. However, the phylogenetic position of Pliobates has been a matter of debates. In this manuscript, new specimens assigned to Pliobates are described and its phylogenetic position is discussed using cladistic analysis. The new material includes lower molars, which were not known in the previous material and clearly show pliopithecoid features such as incomplete pliopithecine triangle. This manuscript provides good evidence to settle the phylogenetic position of Pliobates. Although the present authors' work removes Pliobates from the previously suggested position of a stem hominoid and replaces it within a primitive catarrhine group (pliopithecoids) much more distantly related to us, confirming the pliopithecoid status of Pliobates has nonetheless another interesting influence on the study of hominoid evolution.

The extant hominoids (both gibbons and great apes) are adapted to suspensory locomotion. A simple scenario is that the last common ancestor of gibbons and great apes was already adapted to suspensory locomotion. However, once fossil evidence is considered, the evolution of suspensory adaptations among hominoids seems to be much complicated. Whether suspensory adaptations in hominoids have been evolved independently in various lineages or not has been debated among researchers. This may affect even how we hypothesize the last common ancestor of chimpanzees and humans and how we consider the origin of our own lineage. The results of the present authors' study provide supportive evidence for the hypothesis that suspensory adaptations have been acquired independently in various hominoid lineages.

Accordingly, the contents of this manuscript are quite interesting. However, Nature Communication is not a journal which is highly specialized for primate paleontology. I am afraid that this manuscript, as it is now, is too specialized for most of readers except for a handful of specialists who have direct experience to study pliopithecoid fossils, as background information is not sufficiently provided to readers. Hence, I think it necessary to

revise the manuscript, especially the style of presentation, so that it becomes easier for readers to understand its contents.

For example, there are a number of comparative taxa, many of which are pliopithecoids. The authors give emended diagnosis of *Pliobates* in the main text, and differential diagnosis in the supplementary text, describing differences between *Pliobates* and other taxa.

However, it is difficult for readers to evaluate what the authors write are adequate or not, because there is not sufficient information on the comparative taxa. I think it better to add some more figures, at least in the supplementary information, that include photographs and/or drawings of comparative taxa so that readers can understand more easily the morphological differences that the authors mention in the manuscript.

It would be helpful for readers to add some short explanations in the main text (and more detailed ones in the supplementary information) about the comparative taxa, as well as a figure indicating the chronological and geographical distributions of the comparative taxa. In diagnoses and other parts of the text, the crown proportion (buccolingual index) is often mentioned. Although there is Supplementary Figure 4 (bivariate plots MD X BL), it is better to add another figure, which includes graphs of buccolingual index, so that readers can directly see the differences in BLI between *Pliobates* and other taxa.

In the first description of *Pliobates* (Alba et al., 2015), the cladistic analysis included extant hominoids, extant Old World monkey (*Macaca*), and some fossil hominids such as *Pierolapithecus* and *Hispanopithecus*. Why are they removed in the present cladistic analysis?

Minor comments

L105: "Upper cheek teeth with extensive trigon basin" > "Upper molars..."?

Cheek teeth are premolars and molars. I guess that the authors mean "molars" here.

L116 "metacone and protocone" > "metaconid and protoconid"

Here, the lower p4 morphology is mentioned.

L126-128: How about adding a lack of postprotocristid as another trait of dp4?

L155: "deep trigon basin"

It sounds somewhat odd to use the term trigon basin for premolars when there is not a trigon. In addition, you call the same basin as central fovea in L287. I feel the term central fovea fits better for premolars.

L165: "A small protoconule is..."

In Methods, the authors state that dental nomenclature follows Harrison and Gu (1999). Exactly speaking, Harrison and Gu (1999) use "paraconule" instead of "protoconule".

L211: "trigon basin"

Here, the DP3 morphology is mentioned. Is trigon basin a proper term when there is no distinct metacone or crista obliqua?

L214; "talon basin"

I suppose this is probably a mistypo of "trigon basin".

L232: "distally located protocne"

In non-cercopithecoid catarrhines, the protocone is usually located slightly distally than the paracone. Can it really be a particular trait that connects Pliobates to Barberapithecus and Moergen crouzeline?

L254: "The lingual cingulid is very broad"

I suppose "lingual" is a mistypo of "buccal".

L278: "more derived than Barberapithecus but less so than Plesiopliopithecus and Crouzelia"

I think it better to add some explanatios why the authors think Barberapithecus is more primitive and Plesiopliopithecus and Crouzelia are more derived than Pliobates.

L285: "M1-M2 with paracone not markedly buccally located relative to the metacone" Comapred to other non-cercopithecoid cararrhines, is this really a particular trait that characterize the pliopithecoids?

L298: “entoconid almost transversely aligned with the hypoconid”

Here, this trait is used to distinguish Pliobates from Plesiopliopithecus and Crouzelia (that is, crouzeliines). However, the same trait is used to connect Pliobates to the crouzeliines in L290. Please reconsider these sentences.

L442: What is (30) here?

Reviewer #3 (Remarks to the Author):

In response to the review prompts:

“What are the noteworthy results?” Description of new craniodental material attributed to Pliobates reveals that it has dental traits thought to be diagnostic of membership in Pliopithecoidea. This is interesting in light of its derived wrist and elbow morphology, which had previously suggested stem hominoid status for the genus.

“Will the work be of significance to the field and related fields? How does it compare to the established literature? If the work is not original, please provide relevant references.” It is significant to those working in the area of human evolution, ape evolution, and anthropoid evolution. It adds important evidence to a question that is plagued by missing data.

“Does the work support the conclusions and claims, or is additional evidence needed?”

This is complicated. The cladistic analysis reported in the paper doesn’t really address the important question of where Pliobates fits within anthropoids. It essentially asks what kind of stem catarrhine it is. Its teeth suggest that it is a pliopithecoid, while its postcrania suggest that it may be a stem hominoid. But the cladistic analysis just looks at its teeth in a matrix with very few non-pliopithecoid taxa. Also, the data files supporting the analysis are flawed. Supplementary table 4 is missing a lot of important text.

“Are there any flaws in the data analysis, interpretation and conclusions? Do these prohibit publication or require revision?” At minimum, Supp. Table 4 has to be fixed. It is missing chunks of information.

“Is the methodology sound? Does the work meet the expected standards in your field?” It is not sound if the conclusion it intends to reach is that Pliobates is a pliopithecoid.

“Is there enough detail provided in the methods for the work to be reproduced?” If Supp. Table 4 is completed.

This manuscript presents valuable new information about *Pliobates*. We can now see that its teeth bear traits that suggest that it is a pliopithecoid. However, the cladistic analysis in the manuscript does not fully explore the possible phylogenetic implications of this because it is far too narrow both in terms of character data (being restricted to dental traits) and taxonomic representation. The analysis mostly speaks to the question of what its affinities may be within Pliopithecoidea if it is a pliopithecoid. This is largely irrelevant to the broader significance of the new data, which potentially speak to the question of whether or not *Pliobates* is a pliopithecoid, and consequently whether its ape-like postcranial traits represent convergent acquisitions with respect to crown hominoids. This is the question that makes the new fossils of great importance, and the cladistic analysis presented here doesn't advance that question at all. Unfortunately, the presentation of the analysis is also marred by what I assume are some formatting errors. The character table (Suppl Table 4) is missing chunks of text. This prevented me from examining the character support for various nodes.

To clarify, I oppose the view that every paper about a fossil ape needs to include an exhaustive cladistic analysis. But the potential importance of the new data for *Pliobates* reported here cannot be realized without placing it in a more inclusive analysis. We need to give *Pliobates* (all of its anatomy) the opportunity to cluster with other anthropoids like crown hominoids with which it shares postcranial traits. This complex riddle will only be solved by including as much data as possible. The most critical limiting factor here is anatomical coverage for the fossil taxa, and that is what makes this discovery so very important. It just has to be put to better use. Otherwise, the present analysis could serve well as the basis for a JHE or AJBA paper about the potential phylogenetic placement of *Pliobates* within Pliopithecoidea. But the present analysis does not establish *Pliobates* as a pliopithecoid because it does not test the alternatives.

Below are some specific comments by manuscript line number that may be helpful.

37: If NE is an abbreviation of Northeast, it should be spelled out the first time it is used.

54: I think it is important to think about how this emphasis was manifested methodologically.

56: And of course if they are homoplastic then they are not synapomorphies. Maybe just call them traits of characteristics.

59: Wouldn't the most obvious approach be to investigate its phylogenetic position using an expanded dataset that approaches total evidence as much as possible? You could say that the previous lack of informative dental material for *Pliobates* crippled such an approach, but because of this new discovery it is now possible.

273: The analysis could hardly have found pliopithecoids to be anything else. The cladistic analysis in this manuscript includes only one taxon that is universally considered a crown catarrhine. *Ekembo* is thought by many, perhaps most, to be a very primitive hominoid, but this is by no means unanimous. This is a very good analysis of pliopithecoid phylogenetics, but it is not capable of speaking to the broader issues that make *Pliobates* broadly interesting.

304: Having a combination of primitive and derived traits (which is true of most taxa) does not cause this kind of problem – homoplasy does.

380. I'm afraid the analysis really doesn't speak to this question.

Trait #75 from Supplementary Table 6 isn't in Supplementary Table 4. It is listed as one of the state changes between Node 16 and node 14 in Supplementary Table 6. In fact, something has gone very wrong with Supp. Table 4. Many of the characters are missing, and there are sentence fragments interspersed in various places.

RESPONSE TO REVIEWERS

Ref. NCOMMS-23-22790

Reviewer #1:

This is an excellent study that presents an important reanalysis of the phylogenetic relationships of *Pliobates* from the Late Miocene of Spain. *Pliobates*, one of the best known European Miocene catarrhines, has previously been reported to be a stem hominoid, but the current work provides a re-evaluation of the evidence, including new fossil material, that more convincingly shows it to be a crouzeliid pliopithecoid. The study represents a significant advance in our understanding of Miocene catarrhines in Eurasia. The description of the morphology is detailed and accurate, and the phylogenetic analysis is sound and includes all of the pertinent taxa. I do not have any major critical comments to offer in terms of the reanalysis of the phylogenetic relationships or the general conclusions reached. I recommend publication subject to minor revisions.

Response: We sincerely thank the reviewer for his positive feedback.

There are a few issues to take into consideration before the manuscript is finalized for publication.

(1) The authors contend, at several points in the text, that their finding “support the independent acquisition of postcranial similarities between lesser and great apes”. However, this is not a logical assumption or interpretation, and the claims made here are overstated. Finding parallelisms in an outgroup does not have a direct bearing on the likelihood of homoplasy in the hominoid crown group. One could argue that the independent acquisition of similar morphological features associated with forelimb suspension in multiple unrelated lineages (such as sloth lemurs, atelid primates, and sloths) implies that this could be one plausible evolutionary pathway by which hominoid acquired their specialized postcranial morphology, but it does not represent a valid test of that outcome. In this case, *Pliobates* does not offer evidence that is directly relevant to the question of hominoid postcranial homoplasy - it merely provides a further example of parallel adaptation in a primate as a consequence of similar functional demands.

Response: The reviewer is right that homoplasies between pliopithecoids and crown hominoids do not necessarily imply, *per se*, that these features are not synapomorphic among different crown hominoid lineages. However, our study indicates that purported synapomorphies of crown hominoids independently evolved in another catarrhine lineage, and further highlights the difficulty in distinguishing true homologies from homoplasies caused by convergences among two lineages that are less closely related than hylobatids and hominids (note that the distinction between ‘convergence’ and ‘parallelism’ is ambiguous so we prefer to use the former term only, following Arendt & Reznick, 2008, TREE vol. 23). We think that the more conclusive evidence of postcranial convergences between some pliopithecoids and crown hominoids evinces the plausibility that this might also be the case between hylobatids and hominids, particularly when other lines of evidence from stem hominids are taken into account. Convergences between relatively distantly related primate lineages (such as atelids and hominoids) are more difficult to disentangle than those among more closely related lineages (such as different catarrhine groups). This is illustrated by the previous cladistic results of Alba et al. (2015), which incorrectly recovered *Pliobates* as a stem hominoid because of these homoplasies. Following the reviewer’s concerns, in the revision, we have provided a more nuanced discussion that makes it clear that *Pliobates* does not directly support the convergence hypothesis between hylobatids or hominids, even if shows the plausibility of such a hypothesis. This is further illustrated by means of a second cladistic analysis including also craniodental and postcranial data, which

shows that the inclusion of postcranial features yields similar results for *Pliobates* (confirming its crouzeliid status) but a strikingly different topology for pliopithecoids as a whole, which instead of a clade of stem catarrhines are recovered as a paraphyletic assemblage of stem hominoids (something difficult to believe). Very different results have been also obtained for several Miocene apes (especially *Oreopithecus*) when craniodental and postcranial data are analyzed separately (Pugh, 2022). Therefore, we still consider that our results for *Pliobates* strengthen the view that most parsimonious cladograms of extinct and extant hominoids might be distorted by postcranial homoplasy, to the extent that it could override true phylogenetic signal (see discussion in Urciuoli & Alba, 2023). Said that, we do not contend that all postcranial synapomorphies between hylobatids and hominids are convergences, although we consider that evidence from Miocene hominoids supports the view that this might be the case for many, especially those evolved as an adaptation to suspensory behaviors. We have toned down some of our previous assertions and provided a more nuanced discussion in this regard (based on the new cladistic results), which we hope will be acceptable for the reviewer even if he does not share our views about the potentially misleading effect of postcranial features in hominoid phylogenetics.

(2) The text and tables refer to the crouzeliid from Mörigen. This is an interesting specimen, but the authors initially identify it as an "indeterminate species". However, in the Figures and Tables it is identified as cf. *Pliobates*. This seems to be a taxonomic assessment that requires a more detailed comparative study and further justification. It is particularly problematic because the phylogenetic analysis does not link it as the sister taxon of *Pliobates cataloniae*. It would be best to delete reference to this specimen, since it adds nothing of significance to the overall findings and may be confusing to the general reader.

Response: We only referred to the Mörigen specimen as an indeterminate crouzeliid at the beginning, and at some point in the main text we justified its tentative attribution to *Pliobates*, even if briefly. We acknowledge that the phylogenetic results of the cladistic analysis are not entirely consistent with such a taxonomic attribution, probably because no lower teeth are available, but precisely for this reason we used open nomenclature instead of attributing it to *Pliobates*. The specimen is published and hence we consider that it cannot be ignored in the paper, as it belongs to the same pliopithecoid group as *Pliobates* and closely related taxa. Furthermore, it is an important specimen because it is the only crouzeliine previously known from upper cheek teeth before the find of *Pliobates*. On the other hand, we agree with the reviewer that a more detailed taxonomic assessment, probably including additional analyses and/or material, would be required to better substantiate the assignment of the Mörigen specimen to *Pliobates*. Therefore, following the reviewer's concern, in the revision we have left the specimen unassigned to genus as *Crouzeliinae indet.*

(3) I would recommend that the authors reassess the way in which they analyzed and presented the estimated body mass of *Pliobates*. The authors use the dental metric/body mass allometric equations published by Egi et al. (2004). These regressions and equations are based on species average data, so they provide a reasonable estimation of the average body mass of extinct species. However, applying the equations to estimate the body mass of individual specimens and to use these data to generate intraspecific ranges of body mass or degree of sexual dimorphism may not be valid.

Response: The reviewer is right that these equations are based on mean species data and should be used to estimate average body mass for a species. These equations are the most appropriate for small catarrhines because their computation excluded large primates, but there is no way to circumvent the problem mentioned by the reviewer. Therefore, in the revised version we took a more cautious approach by refraining from estimating body mass separately for different individuals or making any inferences about size dimorphism on this basis. Instead, we estimated body mass based on mean values for each tooth locus, and then we computed

the average value for the various tooth loci available. To reflect this, changes were made in the Morphological comparisons and body mass section of the Results, the Body mass estimation section of the Methods, and the Supplementary Table 3 of the revised version of the manuscript.

(4) The authors state that the “presence of purported crown hominoid synapomorphies” are “functionally related to orthograde behaviors”. I would contend that the synapomorphies relate to orthograde and forelimb suspension, not just orthograde (because the two positional behaviors are not always functionally linked).

Response: We agree, but this seems to be largely a semantic issue because we were implicitly including suspension among orthograde behaviors (indeed, suspensory adaptations independently evolved in atelids were already mentioned a few lines later). The reviewer is apparently equating orthograde behaviors with non-suspensory behaviors, but indeed, orthograde (=antipronograde) behaviors include all those performed with the trunk held vertically (e.g., see recent review in Almécija et al., 2021 Science)—i.e., not only vertical climbing and clambering/bridging, but also brachiation/arm swinging (and bipedalism). Some authors consider that, at least in great apes, orthograde was originally an adaptation to vertical climbing, subsequently co-opted for suspension (see Almécija et al., 2021 Science for further discussion). But this is irrelevant for the sentence alluded to by the reviewer, as we concur that orthograde features are functionally related to both vertical climbing and suspension. We have made this clear in the revision by replacing “functionally related to orthograde behaviors” with “functionally related to orthograde behaviors (i.e., those performed with the trunk held vertically, such as vertical climbing and suspension).”

A few minor corrections and clarifications:

139 & Fig. 3: “Infantile” should read “infant” **Response:** Corrected (also in Supplementary Table 1).

357 & Fig. 6: “Globulous” should read “globular” **Response:** Corrected (also in Supplementary Table 5 of the revised version of the manuscript).

Table 1: The legend needs to explain abbreviation of “BLI” **Response:** Fixed.

736: “Unfinished germ”? **Response:** Replaced with “Incompletely formed germ”.

Fig. 5: Spelling of *Platodontopithecus* and *krishnaii* **Response:** Corrected (also in Supplementary Figure 4 and Supplementary Table 4 of the revised version of the manuscript).

759: “Low zygomatic root” is ambiguous. Better described, perhaps, as “low zygomatic arch root”? **Response:** We respectfully disagree. We are not sure about the ambiguity that the reviewer has in mind, but the zygomatic root is an anatomical landmark customarily designated as such (e.g., Weber & Krenn, 2017 *Anatomical Record*), and adding “arch” is not only unnecessary but might be confusing.

765: Spelling “groove” **Response:** Corrected.
REDACTED

Reviewer #2:

Comments

Pliobates is a small-bodied catarrhine primate discovered from the middle/late Miocene boundary of Spain. In the first description of this genus, it was suggested to be a stem hominoid before the divergence between the hylobatids and hominids, being more derived than the East African stem hominoids (proconsulids). If this is the case, *Pliobates* gives a great influence on our understanding of the last common ancestor of gibbons and great apes. However, the phylogenetic position of *Pliobates* has been a matter of debates.

In this manuscript, new specimens assigned to *Pliobates* are described and its phylogenetic position is discussed using cladistic analysis. The new material includes lower molars, which were not known in the previous material and clearly show pliopithecoid features such as incomplete pliopithecine triangle. This manuscript provides good evidence to settle the phylogenetic position of *Pliobates*. Although the present authors' work removes *Pliobates* from the previously suggested position of a stem hominoid and replaces it within a primitive catarrhine group (pliopithecoids) much more distantly related to us, confirming the pliopithecoid status of *Pliobates* has nonetheless another interesting influence on the study of hominoid evolution.

The extant hominoids (both gibbons and great apes) are adapted to suspensory locomotion. A simple scenario is that the last common ancestor of gibbons and great apes was already adapted to suspensory locomotion. However, once fossil evidence is considered, the evolution of suspensory adaptations among hominoids seems to be much complicated. Whether suspensory adaptations in hominoids have been evolved independently in various lineages or not has been debated among researchers. This may affect even how we hypothesize the last common ancestor of chimpanzees and humans and how we consider the origin of our own lineage. The results of the present authors' study provide supportive evidence for the hypothesis that suspensory adaptations have been acquired independently in various hominoid lineages.

Accordingly, the contents of this manuscript are quite interesting. However, *Nature Communication* is not a journal which is highly specialized for primate paleontology. I am afraid that this manuscript, as it is now, is too specialized for most of readers except for a handful of specialists who have direct experience to study pliopithecoid fossils, as background information is not sufficiently provided to readers. Hence, I think it necessary to revise the manuscript, especially the style of presentation, so that it becomes easier for readers to understand its contents.

Response: We appreciate that the reviewer recognizes, in the second paragraph above, the implications of our study for hominoid evolution despite *Pliobates* being considered a pliopithecoid. That suspensory behaviors evolved independently in several crown hominoids is supported by several lines of evidence, particularly when stem hominids (*Pierolapithecus*) and Miocene pongines (*Sivapithecus*) are considered (see discussion in Alba, 2012 *Evol. Anthropol.*; Almécija et al., 2021 *Science*; Urciuoli & Alba, 2023 *JHE*). *Pliobates* contributes to this debate by showing that some postcranial features shared by extant hominoids independently evolved in another catarrhine lineage. As noted in one of our responses to reviewer 1 above, at the very least, this highlights the difficulties in distinguishing between homologies and homoplasies in these features among closely related lineages—and, when considered with evidence from stem hominids, evince that similar convergences are plausible between hylobatids and hominids.

We also appreciate the reviewers' constructive criticism that the paper should be revised to make it more readily accessible to a wider audience. The reviewer considers that the problem to make the paper more readily available to other researchers is the lack of sufficient "background" because it seems too specialized. We need to provide the details of dental morphology in which our analyses are based but, as explained in greater detail below, we have done our best to improve this aspect along the lines suggested by the reviewer. Given the space constraints of the journal, this is more easily said than done, but we hope that the reviewer will find the manuscript importantly improved in this regard.

For example, there are a number of comparative taxa, many of which are pliopithecoids. The authors give emended diagnosis of *Pliobates* in the main text, and differential diagnosis in the supplementary text, describing differences between *Pliobates* and other taxa. However, it is difficult for readers to evaluate what the authors write are adequate or not, because there is not sufficient information on the comparative taxa. I think it better to add some more figures, at least in the supplementary information, that include photographs and/or drawings of

comparative taxa so that readers can understand more easily the morphological differences that the authors mention in the manuscript.

Response: Given the number of tooth positions described for *Pliobates*, it would be unreasonable to provide comparative figures for all of them (this, indeed, would be something more suitable for a specialized paleoprimatological journal). However, following the reviewer's request we have added to the supplementary information three figures depicting the morphology of the lower molars and fourth deciduous premolar of *Pliobates* as compared with other pliopithecoids (Supplementary Figs. 1–3 in the revised version of the manuscript). These are the teeth that most clearly display crouzeliid affinities and, hence should be the most helpful for readers. Unfortunately, we do not have original pictures of most of the comparative material, which were otherwise depicted based on casts (which we think is acceptable for the supplementary information but not for the main text).

It would be helpful for readers to add some short explanations in the main text (and more detailed ones in the supplementary information) about the comparative taxa, as well as a figure indicating the chronological and geographical distributions of the comparative taxa.

Response: The information that can be provided in the main text is very restricted due to the space constraints of the journal, especially considering that we have expanded the text to accommodate a second cladistic analysis in response to reviewer 3 (see below). Furthermore, contextual information (age and geographic provenance) is not normally provided in diagnoses. However, we have briefly added such information (list of the taxa investigated with provenance) when mentioning comparative taxa in the Cladistic analysis section of the Methods. Furthermore, in the supplementary information, we have added a new table (Supplementary Table 6 in the revised version of the manuscript) providing the taxonomic categories (superfamily and family), chronostratigraphic range, geographical distribution, and types of remains preserved (dental, cranial, and/or postcranial remains), along with the main references, for all the taxa in the comparative sample. We believe that this information is better conveyed in a table rather than in a figure (as suggested by the reviewer), since it seems quite difficult to synthesize geographic distribution and chronostratigraphic ranges into a single figure. Nevertheless, we have further incorporated the stratigraphic range of each species in the cladogram based on dental remains.

In diagnoses and other parts of the text, the crown proportion (buccolingual index) is often mentioned. Although there is Supplementary Figure 4 (bivariate plots MD X BL), it is better to add another figure, which includes graphs of buccolingual index, so that readers can directly see the differences in BLI between *Pliobates* and other taxa.

Response: We assume the reviewer refers to the breadth/length index (not "buccolingual" index). Following their request, we have added a new supplementary figure (Supplementary Figure 5 in the revised version) that depicts boxplots for the BLI.

In the first description of *Pliobates* (Alba et al., 2015), the cladistic analysis included extant hominoids, extant Old World monkey (*Macaca*), and some fossil hominoids such as *Pierolapithecus* and *Hispanopithecus*. Why are they removed in the present cladistic analysis?

Response: Because, as stated in the manuscript, the aim of the present analysis was to clarify the phylogenetic relationships of *Pliobates* among pliopithecoids, as it is obvious from the newly described teeth that it is a crouzeliid. Given that the previous analysis by Alba et al. was misguided by the postcranial convergences with hominoids and that other cladistic analyses subsequently obtained different results, we thought it was not necessary to replicate Alba et al.'s analysis with the newly described teeth. Indeed, this is something we aimed to do in the future. Nevertheless, as this concern was also raised by reviewer 3, we added a second cladistic analysis to the revised version (see our responses below to reviewer 3 for

further details). Some of the reviewer comments (see below) about particular dental traits made us carefully revise the original data matrix. We noticed some minor coding and/or character definition mistakes in the original data that do not result in any change in the tree topology but give slightly different tree score and bootstrap/index values. The manuscript has been modified accordingly. The changes introduced in character scoring relative to the previous version are the following:

#10: (0) instead of (1) for *Ekembo*.

#31: (0) instead of (0,1) for *Ekembo*.

#35: (0) instead of (1) for *Aegyptopithecus*, *Ekembo*, and *Victoriapithecus*, and (1) instead of (2) for pliopithecoids.

#53: (1) instead of (0) for *P. zhanxiangi* and *Krishnapithecus*.

#69: (0,1) instead of (0) for *Victoriapithecus*.

The changes introduced in character definition are:

#24: (1) should be "subtriangular to slightly rhomboid".

#35: (0) should be "long" and (1) "short".

#66: state definitions should be inverted.

#76: (2) should be "narrow to absent" instead of "narrow".

Minor comments

L105: "Upper cheek teeth with extensive trigon basin" > "Upper moars...?"

Cheek teeth are premolars and molars. I guess that the authors mean "molars" here.

Response: No, we meant all the upper cheek teeth, but we concur with the reviewer that the term "trigon basin" cannot be applied to the upper premolars because they lack a metacone (and, hence, have no trigon). For the upper premolars, Harrison and Gu (1999) used the term "central fovea" to refer to the basin delimited by the hypoparacrista (mesially) and the hypoprotocrista (distally), resulting in a distal fovea mesially delimited by the hypoprotocrista. However, when the hypoprotocrista is lacking, no "central" and "distal" foveae can be distinguished, in which case Harrison and Gu (1999) employed instead the term "distal basin" to refer to the large fovea delimited mesially by the hypoparacrista. Because *Pliobates* lacks a hypoprotocrista, we replaced "trigon basin" with "distal basin" when referring to the upper premolars.

L116 "metacone and protocone" > "metaconid and protoconid"

Here, the lower p4 morphology is mentioned.

Response: Corrected.

L126-128: How about adding a lack of postprotocristid as another trait of dp4?

Response: We disagree with this interpretation, as it is implicit from the text alluded to by the reviewer: "a more obliquely oriented cristid obliqua (directed toward the hypoprotocristid and hypometacristid junction)". The reviewer is assuming that the postprotocristid has been lost and that the cristid obliquid is exclusively constituted by the prehypocristid. But the cristid is normally constituted by the merging of the prehypocristid and the postprotocristid, and our interpretation is that in this taxon (and other crouzeliids with the same occlusal pattern), the origin of the postprotocristid has been displaced from the distal aspect of the protoconid toward the hypoprotocristid. Hence, the addition suggested by the reviewer is unnecessary, as it is implicit in our sentence. To avoid misunderstandings, in the revised version, we have been more explicit in this regard in the main text: "a more obliquely oriented cristid obliqua (directed toward the hypoprotocristid and hypometacristid junction, so that the postprotocristid originates from the latter junction)". We have also been more explicit in this regard in the supplementary information.

L155: "deep trigon basin"

It sounds somewhat odd to use the term trigon basin for premolars when there is not a trigon. In addition, you call the same basin as central fovea in L287. I feel the term central fovea fits better for premolars.

Response: We agree with the reviewer and, as explained above in a previous response to the reviewer, we replaced this term with "distal basin" throughout the manuscript when referring to the upper premolars instead of molars.

L165: "A small protoconule is..."

In Methods, the authors state that dental nomenclature follows Harrison and Gu (1999). Exactly speaking, Harrison and Gu (1999) use "paraconule" instead of "protoconule".

Response: The reviewer is right regarding Harrison & Gu (1999), so we changed the term in the revised version. However, both terms are synonyms, as noted in Alba et al. (2010) AJPA and Alba et al. (2013) JHE, among others, where "protoconule" was used instead, following Swinder (2002).

L211: "trigon basin"

Here, the DP3 morphology is mentioned. Is trigon basin a proper term when there is no distinct metacone or crista obliqua?

Response: We concur with the reviewer that there is no trigon due to the lack of a distinct metacone (it is very rudimentary). Because there is no crista obliqua separating a central fovea from a distal fovea, we followed the same rationale as for the upper premolars (see above) and replaced the term "trigon basin" with "distal basin" in the revised version.

L214; "talon basin"

I suppose this is probably a mistypo of "trigon basin".

Response: Yes, thanks for catching this. Corrected in the revised version.

L232: "distally located protocone"

In non-cercopithecoid catarrhines, the protocone is usually located slightly distally than the paracone. Can it really be a particular trait that connects *Pliobates* to *Barberapithecus* and Moergen crouzeline?

Response: The reviewer is right. However, our comparisons indicate differences in this regard among non-cercopithecoid catarrhines: in crouzeliids, *Pliopithecus*, *Aegyptopithecus* and *Saadanius*, the protocone is more distally located than in dionysopithecids, dendropithecids, and *Ekembo* (see also Ji et al., 2022). This is not a trait exclusively shared by *Pliobates*, *Barberapithecus*, and the crouzeliine from Mörge but just one of the features that they have in common. The same could be said, for example, of the relatively small metacone, which also occurs in dendropithecids, pliopithecids, *Dionysopithecus*, and the crouzeliid *Fanchangia*, but not in other pliopithecoids and stem catarrhines, or the crown catarrhines investigated. Our sentence was aimed to stress that *Pliobates* more closely resembles *Barberapithecus* and the crouzeliine from Mörge in the degree of molar cups compression and crest sharpness but that it also shares with them other occlusal details (listed within parentheses) without claiming that they are exclusively displayed by them. To avoid potential misunderstandings, we replaced "distally located protocone" with "more distal location of the protocone, i.e., more so than in other taxa".

L254: "The lingual cingulid is very broad"

I suppose "lingual" is a mistypo of "buccal".

Response: Yes, fixed in the revised version.

L278: "more derived than *Barberapithecus* but less so than *Plesiopliopithecus* and *Crouzelia*"
I think it better to add some explanation why the authors think *Barberapithecus* is more primitive and *Plesiopliopithecus* and *Crouzelia* are more derived than *Pliobates*.

Response: This fragment referred to the branching order of these taxa in the cladogram recovered by our cladistic analysis, as it is obvious from the sentence as a whole: "*Pliobates* is recovered as a crouzeliine, more derived than *Barberapithecus* but less so than *Plesiopliopithecus* and *Crouzelia*, which are recovered as sister taxa." To avoid misunderstandings, the sentence was rephrased as "less basal than *Barberapithecus* but more so than *Plesiopliopithecus* and *Crouzelia*" Note, however, that what the reviewer is requesting was already stated in the next paragraph already in the previous version of the manuscript (L291-295): "Finally, among crouzeliids, *Pliobates* most closely resembles *Plesiopliopithecus* and *Crouzelia* in the possession of multiple synapomorphies (female c1 relatively broad with a cuspid-like mesiolingual cristid-lingual cingulid; molars with and markedly sharp crests; very narrow m1; hypoprotocristid originating distally from protoconid with inclined cristid oblique connecting to where the hypoprotocristid originates, at least in m1; and distal fovea not separated from the talonid basin, at least in m2)". To be more clear-cut in this regard, in the revised version we added "absent from *Barberapithecus*" after "multiple synapomorphies".

L285: "M1-M2 with paracone not markedly buccally located relative to the metacone"
Compared to other non-cercopithecoid cararrhines, is this really a particular trait that characterizes the pliopithecoids?

Response: Based on our observations, the paracone is indeed more markedly buccally located (relative to the metacone) in dendropithecids as well as *Aegyptopithecus* and *Saadanius*, especially in the M2. For that reason, we codified these taxa differently than pliopithecoids in the matrix.

L298: "entoconid almost transversely aligned with the hypoconid"

Here, this trait is used to distinguish *Pliobates* from *Plesiopliopithecus* and *Crouzelia* (that is, crouzeliines). However, the same trait is used to connect *Pliobates* to the crouzeliines in L290. Please reconsider these sentences.

Response: Based on our cladistic analysis, in the present paper we consider crouzeliines as including *Barberapithecus*, the Mörigen crouzeliine, *Pliobates*, *Plesiopliopithecus*, and *Crouzelia*, not just *Plesiopliopithecus* and *Crouzelia*. The character that corresponds to the trait mentioned by the reviewer is #084: m1–m2 entoconid distal position relative to the hypoconid: 0, clearly more distal; 1, almost transversely aligned. State (1) is inferred by the analysis to be an ambiguous synapomorphy of crouzeliines, present in *Barberapithecus* and *Pliobates* but absent in *Plesiopliopithecus* and *Crouzelia*, which display state (0), interpreted as a reversal. To be more specific, when listing the traits that link *Pliobates* to crouzeliines, we replaced "entoconid almost transversely aligned with hypoconid" with "entoconid almost transversely aligned with hypoconid (even though this synapomorphy is ambiguous because it would have been reversed in *Plesiopliopithecus* and *Crouzelia*, which display an entoconid clearly more distal than the hypoconid)".

L442: What is (30) here?

Response: We thank the reviewer for spotting this. It is an inadvertent mistake. The number of the reference cited should be (1), fixed in the revision.

Reviewer #3:

In response to the review prompts:

"What are the noteworthy results?" Description of new craniodental material attributed to *Pliobates* reveals that it has dental traits thought to be diagnostic of membership in

Pliopithecoidea. This is interesting in light of its derived wrist and elbow morphology, which had previously suggested stem hominoid status for the genus.

Response: This is a fair account of our contribution, except that “thought to be diagnostic” implies they are no longer diagnostic, which is not the case. Some of the dental traits revealed by the new material are still diagnostic of pliopithecoids, and some of crouzeliids and crouzeliines, as confirmed by our previous cladistic analysis.

“Will the work be of significance to the field and related fields? How does it compare to the established literature? If the work is not original, please provide relevant references.” It is significant to those working in the area of human evolution, ape evolution, and anthropoid evolution. It adds important evidence to a question that is plagued by missing data.

Response: We agree.

“Does the work support the conclusions and claims, or is additional evidence needed?” This is complicated. The cladistic analysis reported in the paper doesn’t really address the important question of where *Pliobates* fits within anthropoids. It essentially asks what kind of stem catarrhine it is. Its teeth suggest that it is a pliopithecoid, while its postcrania suggest that it may be a stem hominoid. But the cladistic analysis just looks at its teeth in a matrix with very few non-pliopithecoid taxa. Also, the data files supporting the analysis are flawed. Supplementary table 4 is missing a lot of important text.

Response: As clarified above by the editor’s comments, the issue with the files was already fixed (we apologize for the inconvenience). As for the reviewer’s contention that “the paper doesn’t really address the important question of where *Pliobates* fits within anthropoids”, we strongly disagree as we clearly asserted that *Pliobates* is a crouzeliine pliopithecoid, and the pliopithecoid status was even noted in the title. The reviewer is implicitly assuming that providing a cladogram is the only valid way to demonstrate that a given taxon is what morphology shows it to be. This can sometimes be of use, and it can also be misleading, as it clearly was in Alba et al. (2015), as the crouzeliid synapomorphies shown by the newly reported teeth of *Pliobates* unambiguously indicate that it is a pliopithecoid. We are not arguing that dental features are not subject to homoplasy, but clearly, dental morphology is the most reliable source of information for deciphering the taxonomic affinities of extinct primates. In this case, *Pliobates* shows some features that have been only found among pliopithecoids (such as the distal arm of the pliopithecine triangle), whereas postcranial convergences have been previously noted between crown hominoids, atelids, and even other taxa such as lorids. This is further discussed below in response to additional comments by the reviewer in this regard, but to make a long story short, in the revised version, we also performed a second cladistic analysis that includes a wider representation of taxa as well as cranial and postcranial features. Nevertheless, for the reasons explained below, we also analyzed separately two partitions of the data, as the postcranium is clearly more affected by functional than true phylogenetic signal. Regarding Supplementary Table 4 the file has been uploaded again (Excel file) as Supplementary Data 1.

“Are there any flaws in the data analysis, interpretation and conclusions? Do these prohibit publication or require revision?” At minimum, Supp. Table 4 has to be fixed. It is missing chunks of information.

Response: As explained above, this was already fixed by the editor in the previous round. The data were reuploaded and correctly labeled this time.

“Is the methodology sound? Does the work meet the expected standards in your field?” It is not sound if the conclusion it intends to reach is that *Pliobates* is a pliopithecoid.

Response: We strongly disagree with the reviewer’s assessment in this regard, for the reasons anticipated above and explained in greater detail below. No cladistic analysis would be necessary to show this, and we already provided one that shows *Pliobates* deeply embedded

within crouzeliid pliopithecoids based on the most reliable source of taxonomic information: teeth. Nevertheless, the broader cladistic analysis requested by the reviewer has been provided in the revised version.

“Is there enough detail provided in the methods for the work to be reproduced?” If Suppl. Table 4 is completed.

Response: As noted above, this concern no longer applies.

This manuscript presents valuable new information about *Pliobates*. We can now see that its teeth bear traits that suggest that it is a pliopithecoid. However, the cladistic analysis in the manuscript does not fully explore the possible phylogenetic implications of this because it is far too narrow both in terms of character data (being restricted to dental traits) and taxonomic representation. The analysis mostly speaks to the question of what its affinities may be within Pliopithecoidae if it is a pliopithecoid. This is largely irrelevant to the broader significance of the new data, which potentially speak to the question of whether or not *Pliobates* is a pliopithecoid, and consequently whether its ape-like postcranial traits represent convergent acquisitions with respect to crown hominoids. This is the question that makes the new fossils of great importance, and the cladistic analysis presented here doesn't advance that question at all. Unfortunately, the presentation of the analysis is also marred by what I assume are some formatting errors. The character table (Suppl Table 4) is missing chunks of text. This prevented me from examining the character support for various nodes.

Response: We appreciate that the reviewer understands the importance of the fossils described in our manuscript but respectfully disagree with their criticisms in the paragraph above. The newly described lower teeth do not merely “suggest” *Pliobates* is a pliopithecoid. As noted in a previous response to this reviewer above, they clearly show this to be the case, to the extent that no cladistic analysis would be required to make the point that *Pliobates* is a pliopithecoid of the family Crouzeliidae. To say that our cladistic analysis “doesn't advance that question at all” is a most unfair criticism when it is taken into account that our cladistic results are the best-resolved phylogeny of pliopithecoids published so far and unambiguously recover *Pliobates* as a crouzeliine. The reviewer is missing the point that our cladistic analysis of dental traits (the only one provided in the previous version of the manuscript) was explicitly intended to clarify the phylogenetic relationships of *Pliobates* with other crouzeliids, not to confirm its pliopithecoid affinities. In other words, the differential diagnosis would have been enough to demonstrate that it belongs to this pliopithecoid family, but our cladistic analysis was required to pinpoint to what crouzeliids it is more closely related.

The reviewer implicitly questions the taxonomic assessment that *Pliobates* is a crouzeliid pliopithecoid and assumes that a cladistic analysis including also cranial and postcranial features, along with a broader representation of hominoids, might still indicate that *Pliobates* is a stem hominoid, as in Alba et al. (2015). Given the crouzeliid synapomorphies displayed by the lower teeth of *Pliobates* as well as the results of our dental-only cladistic analysis, we disagree with such an interpretation, which entirely ignores the problem that *Pliobates* highlights regarding cladistic analysis, including postcranial data: namely, that there is a lot of postcranial homoplasy (at least, between *Pliobates* and crown hominoids) distorting the cladistic results. Probably, this would not happen if there were less missing data (for many fossil hominoid and pliopithecoid taxa). In any case, the bias introduced by postcranial homoplasy has been recently restated by Pugh (2022) JHE and even more explicitly by Urciuoli & Alba (2023) JHE, based on the contradictory results obtained by craniodental and postcranial data separately. As noted by the latter authors, a problem of long-branch attraction caused by postcranial convergences between hylobatids and hominids might be distorting the phylogenetic analyses of crown hominoids, and *Pliobates* indirectly supports the plausibility of this hypothesis by showing the distorting effect that such homoplasies had in Alba et al.'s (2015) cladogram. In other words, the postcranial features the reviewer wants us to include in the cladistic analysis are the main reason (coupled with the fact that lower teeth were previously unknown for *Pliobates*) why the previous cladistic

analysis by Alba et al. (2015) recovered *Pliobates* as a hominoid. Other cladistic analyses, including additional taxa and using a different character selection (with less emphasis on these postcranial features), already recovered *Pliobates* as a pliopithecoid (Nengo et al., 2017 Nature; Gilbert et al., 2020 PRSB). Sincerely, we do not understand why the reviewer remains so skeptical and considers the pliopithecoid affinities of *Pliobates* to be unsettled after the more conclusive dental evidence provided in our manuscript. REDACTED

Said that, although we disagree with the validity of the reviewer's criticism in this regard, their request is not unreasonable. So, we decided to address it in the revision by performing a second cladistic analysis that includes a wider representation of taxa as well as craniodental and postcranial characters. This analysis was based on the original matrix by Alba et al. (2015), as this is the only one that has thus far recovered *Pliobates* as a hominoid, but updated based on the dental characters coded in our manuscript, coupled with some additional cranial features based on recently published literature. As a disclaimer, it should be noted that performing a total evidence (i.e., craniodental + postcranial) cladistic analysis of *Pliobates* was outside the scope of this paper. Therefore, until ongoing research is completed, for the cranial and postcranial characters, we had to mostly rely on the previous matrix of Alba et al. (2015), although with some improvements. We refrained from including species from the dental cladistic analysis that were not included in Alba et al. (2015), because their large number of missing data for cranial and postcranial features makes their position unstable and would collapse most of the clades. The cranial and postcranial characters included were updated according to the progress made in the field since 2015, for example, by replacing *Proconsul* s.l. by *Ekembo* (i.e., excluding *Proconsul* s.s.). We also grouped all *Pliopithecus* species (as in Alba et al., 2015) to perform the analysis at the genus level. However, unlike in the latter paper, we kept separate the dendropithecids *Simiolus* and *Dendropithecus*, as well as the crouzeliines *Crouzelia* and *Plesiopliopithecus*. Moreover, besides updating the dental characters based on the present paper, we added characters based on the inner ear semicircular canals and improved the coding of some other cranial and postcranial features based on recently published papers. Details on character modifications are given in Supplementary Methods of the revised version.

The results of the new cladistic analyses (Supplementary Fig. 6 in the revised version) not only support our previous assessment that *Pliobates* is a crouzeliid pliopithecoid but further confirm our suspicions about the confounding effect of postcranial homoplasies. As explained above, in all analyses considering (cranio)dental characters *Pliobates* branches off among other crouzeliids but, when postcranial characters are included, all pliopithecoids are recovered as a paraphyletic grouping of stem hominoids instead of stem catarrhine clade. In the postcranial-only analysis, all the clades are collapsed except crown hominoids (i.e., excluding *Ekembo*), and *Pliobates* is recovered as a stem hominoid. The reviewer is free to favor the results based on all evidence available, but considering the other analyses and especially the discrepancies between craniodental and postcranial analyses, our interpretation is that postcranial convergences between *Pliobates* and hominoids are misleading the most parsimonious cladogram due to the wrong signal introduced by postcranial homoplasies. And, in any case, even if pliopithecoids were stem hominoids, this would not invalidate our assessment (as confirmed by the new cladistic analyses) that *Pliobates* is a crouzeliid.

To clarify, I oppose the view that every paper about a fossil ape needs to include an exhaustive cladistic analysis. But the potential importance of the new data for *Pliobates* reported here cannot be realized without placing it in a more inclusive analysis. We need to give *Pliobates* (all of its anatomy) the opportunity to cluster with other anthropoids like crown hominoids with which it shares postcranial traits. This complex riddle will only be solved by including as much data as possible. The most critical limiting factor here is anatomical coverage for the fossil taxa, and that is what makes this discovery so very important. It just has to be put to better use. Otherwise, the present analysis could serve well as the basis for a JHE or AJBA paper about the

potential phylogenetic placement of *Pliobates* within Pliopithecoidea. But the present analysis does not establish *Pliobates* as a pliopithecoid because it does not test the alternatives.

Response: The reviewer's assertion contradicts their request to provide a more comprehensive cladogram in the present manuscript. In addition, the reviewer contends that the systematic position of *Pliobates* can only be solved by including as much data as possible. However, we consider that postcranial data (due to abundant homoplasy) is the main factor that misled the results of Alba et al. (2015). The abundant missing data for many taxa (a problem noted by the reviewer), coupled with abundant postcranial morphology, is arguably distorting the results. Nevertheless, as already noted in the previous response to this reviewer, we decided to give *Pliobates* the chance to cluster with other anthropoids, while at the same time analyzing the craniodental and postcranial features separately, and the results based on all evidence available support its crouzeliid status but challenge instead the stem catarrhine status of pliopithecoids as a whole.

We would like to stress that we do not consider the reviewer's request unreasonable (this is why we fulfilled it), we just disagree that such analysis was necessary to conclusively show that *Pliobates* is a pliopithecoid. Performing a more comprehensive analysis of *Pliobates* not restricted to dental characters is something that we had been planning for years. At first, we regretted to be forced to do so at this time, when there is plenty of ongoing research about *Pliobates* that will hopefully enable to code additional cranial and postcranial characters of this taxon. Nevertheless, we now realize this was worth the effort, because our new cladistic results more clearly evince the confounding effect of postcranial homoplasy without contradicting the pliopithecoid status of *Pliobates*. We therefore think that, thanks to the reviewer's insistence, the revised version is much more solid in this regard. Both the Results and Discussion have been reorganized and rewritten to an important extent to accommodate space for the new cladistic results. The Methods (Cladistic analysis section) now include a small paragraph on the new cladistic analyses. The Abstract and Introduction were also modified accordingly. As specified above, the Supplementary Methods provide details on how the new analyses were performed. The Supplementary Data 3 and 4 provided along with the revised version of the manuscript correspond to the new character definitions and the new matrix, respectively. Supplementary Text 4 discusses in greater detail the new results that are shown in Supplementary Fig. 6.

Below are some specific comments by manuscript line number that may be helpful.

37: If NE is an abbreviation of Northeast, it should be spelled out the first time it is used. **Response:** Done, but we see no reason to capitalize "northeast".

54: I think it is important to think about how this emphasis was manifested methodologically.

Response: We are not sure what the reviewer implies with this comment. There is no objective way to determine into how many discrete characters the morphology of a particular bone must be subdivided. There are more objective ways to discretize continuous features, but not to determine how many characters best represent a particular structure. This is indeed one of the main reasons as to why there is an inherent subjectivity in cladistic analyses, the characters coded by different authors will largely depend on their expertise and other unconscious biases, and even those suspected to be homoplastic should not be removed, as (from a cladistic epistemological viewpoint) homoplasy can only be determined a posteriori based on the most parsimonious topology. An analysis of all the evidence available is always preferable on epistemological terms, but the outcome will be determined by the number of characters included from different areas that, for whatever reason, provide a different signal than others (see discussion in Urciuoli & Alba, 2023 JHE). Given this caveat, we consider that the most suitable approach is to be as clear as possible in this regard by following the example of Pugh (2022), who separately analyzed craniodental and postcranial features to detect the inconsistent topologies recovered for some taxa. Our interpretation of the results has already

been outlined in previous responses to this reviewer and is exposed in greater detail in the revised Discussion.

56: And of course if they are homoplastic then they are not synapomorphies. Maybe just call them traits of characteristics.

Response: We understand the reviewer's viewpoint but consider that is preferable to just add "purported" before "synapomorphies".

59: Wouldn't the most obvious approach be to investigate its phylogenetic position using an expanded dataset that approaches total evidence as much as possible? You could say that the previous lack of informative dental material for *Pliobates* crippled such an approach, but because of this new discovery it is now possible.

Response: We disagree, for the reasons explained above, namely: (1) an expanded dataset would preclude analyzing the internal phylogeny of pliopithecoids because many of them have a lot of missing data regarding cranial and postcranial features; (2) there are strong reasons to suspect that the possession of homoplastic postcranial features led Alba et al.'s (2015) cladistic analysis to recover *Pliobates* as a hominoid. Nevertheless, as also explained above, the reviewer's request is reasonable, and hence we added a second cladistic analysis based on an extended dataset with additional taxa and characters, as requested by the reviewer. See above for an outline of the new cladistic results, which confirm the crouzeliid status of *Pliobates* and illustrate the misleading effects of postcranial convergences.

273: The analysis could hardly have found pliopithecoids to be anything else. The cladistic analysis in this manuscript includes only one taxon that is universally considered a crown catarrhine. *Ekembo* is thought by many, perhaps most, to be a very primitive hominoid, but this is by no means unanimous. This is a very good analysis of pliopithecoid phylogenetics, but it is not capable of speaking to the broader issues that make *Pliobates* broadly interesting.

Response: We think that's not the case, for two good reasons. First, we see no reason as to why our dental cladistic analysis could not have recovered pliopithecoids as the sister taxon of dendropithecids instead of as a more basal lineage of stem catarrhines, or even as stem hominoids more closely related to *Ekembo*, if the character scoring for *Pliobates* and other taxa would have been different. Second, we are not sure who the reviewer has in mind when asserting that not everyone agrees that *Ekembo* is a stem hominoid REDACTED, but the question here is that Alba et al. (2015) not only recovered *Pliobates* as a stem hominoid, but also as more derived than *Ekembo*, which is contradicted by the cladistic analysis provided in the previous version of the manuscript. In any case, this criticism no longer applies after the new cladistic analysis included in the revised version.

304: Having a combination of primitive and derived traits (which is true of most taxa) does not cause this kind of problem – homoplasy does.

Response: The reviewer is misrepresenting our original sentence, which specified a "mosaic of primitive (stem catarrhine-like) and derived (hominoid-like) features, particularly in the postcranium". In other words, *Pliobates* displays some derived features of crown hominoids while lacking features derived of crown catarrhines. A stem hominoid would be expected to display a mixture of primitive and derived features compared with crown hominoids, but not compared to crown catarrhines as a whole. Of course, this is caused by homoplasy, but we believe this is just another (and more vague) way to put it and that our original sentence is clear enough to keep it—particularly given the aim of the sentence, which was to explain the discrepancy between different cladistic analyses depending on the emphasis put on the primitive features or on the hominoid-like ones. No changes were thus introduced in this particular sentence.

380. I'm afraid the analysis really doesn't speak to this question.

Response: This comment is difficult to understand. The sentence alluded reads "Deciphering the phylogenetic relationships between pliopithecoids, crown catarrhines, and other Miocene catarrhines is not the aim of this paper". So, of course the analysis provided in the previous version of the manuscript did not speak about this question. We felt compelled to prevent people from thinking otherwise, but either the reviewer misread the sentence to mean just the opposite or we misinterpreted their comment as a criticism when indeed the reviewer was agreeing with our passage. In any case, after the addition of a new, more comprehensive cladistic analysis, the revised manuscript does more directly tackle this issue, and hence we modified the sentence to make it more clear: "Deciphering the phylogenetic relationships among pliopithecoids, crown catarrhines, and other Miocene catarrhines is not the main aim of this work but the results of our cladistic analyses support the view that pliopithecoids (including Pliobates) are stem catarrhines and that previous results indicating a stem hominoid status are probably attributable to the independent acquisition in crouzeiliids of postcranial ape-like features".

Trait #75 from Supplementary Table 6 isn't in Supplementary Table 4. It is listed as one of the state changes between Node 16 and node 14 in Supplementary Table 6.

In fact, something has gone very wrong with Supp. Table 4. Many of the characters are missing, and there are sentence fragments interspersed in various places.

Response: We apologize for the inconvenience but don't understand what happened exactly, as the files were correctly formatted (provided as Excel files) and the problem according to the Editor is that we incorrectly uploaded them as Supplementary Information instead of Supplementary Data. Anyway, our mistake was already fixed by the Editor during the previous round and, when submitting the revision, we ensured that the files were correctly uploaded as Supplementary Data (1 and 2, respectively) as Word files.

REVIEWERS' COMMENTS

Reviewer #1 (Remarks to the Author):

The authors have adequately addressed the issues that were raised in the reviews and I recommend that the revised version be accepted for publication.

A few minor wrinkles that need to be resolved prior to publication:

L33: Scholars working on hylobatids object to the use of the term "lesser ape" because of its

pejorative connotation. It would be better to use hylobatids instead.

L54: "in" should read "on"

L125-126 + L221: "very mesially tapering" would be better as "mesially very tapering"

L232: Delete "entirely". Redundant.

L348: "resolutive" is an odd legal term to use here. Rephrase?

L381: Delete hyphen between "stem" and "catarrhine"

L458: "true phylogenetic signal". It's problematic to invoke phylogenetic "truth" in regard to

fossil taxa. Rephrase?

L461: "along" should read "among"

L536: Spelling of "macinnesi"

Reviewer #2 (Remarks to the Author):

As for my previous comments to the manuscript, I think they have been improved well in this revised version. The photographs of comparative specimens added in the supplementary figures are nice and useful both for general readers and specialists. I think this manuscript is worth to be published with a few minor corrections (see below).

Supplementary Figure 5

Check the calculation for the upper molars. The index may have been calculated as length/breadth (%), instead of breadth/length. As the upper molars are buccolingually broader than mesiodistally long, the breadth/length indices should be over 100%.

Supplementary Text 4.

“Alternatively, it may be argued that that the total evidence results are” >

“Alternatively, it may be argued that the total evidence results are....”

(Remove the unnecessary “that”.)

Reviewer #3 (Remarks to the Author):

The authors have addressed my principal criticism by adding an additional cladistic analysis that demonstrates the important point they were trying to make about postcranial homoplasy.
REDACTED

The authors agree that my requested broader cladistic analysis is “reasonable”, a word they chose, but only after repeatedly arguing that it is not necessary and that I’m only requesting it because I don’t believe their attribution of Pliobates to the crouzeliid family of pliopithecoids. There are three things I want to say about this.

The first is that I don’t see why they think it is a reasonable suggestion if my reasons for requesting it are wrong. Wouldn’t that make it literally unreasonable – without reason? The second is that I think their objections are unreasonable. What they seem to be saying is that they can tell this species is a crouzeliid just by looking at it (based on its diagnostic traits), and only need a cladistic analysis to tell them more precisely what kind of pliopithecoid it is. Imagine if I did that with Oreopithecus and a selection of nyanzapithecines along with Ekembo, Saadania, and Aegyptopithecus. I would find out just what kind of nyanzapitheine Oreopithecus is. I don’t think that would convince Reviewer #1 that it is a nyanzapithecine (or technically that they are all oreopithecids). I could expand greatly on why I think their objections were misguided, but there seems little point because they took my suggestion anyway.

The third is this – I think the paper is much better for having added the broader cladistic analysis. I think it makes one of their important points more clearly than before.

Regarding the novelty of the information provided in this manuscript, it is my opinion that most of what is important about Pliobates was already published in the 2015 Science paper. It is true that the conclusions of the two papers are quite different because that one placed Pliobates as a stem hominoid, but the fact is that many of us (primate paleontologists who study and have published papers about catarrhines including pliopithecoids) did not think that it was a hominoid then based on the teeth already known. True, they did not exhibit any diagnostic pliopithecoid traits, but they were very unlike the teeth of crown hominoids. In that light, the refinement of its phylogenetic position offered here is significant, but I don’t think it justifies publication in a high impact journal again.

REDACTED

Here and below, my response to their response is in green.

In response to the review prompts:

“What are the noteworthy results?” Description of new craniodental material attributed to *Pliobates* reveals that it has dental traits thought to be diagnostic of membership in Pliopithecoidea. This is interesting in light of its derived wrist and elbow morphology, which had previously suggested stem hominoid status for the genus.

Response: This is a fair account of our contribution, except that “thought to be diagnostic” implies they are no longer diagnostic, which is not the case. Some of the dental traits revealed by the new material are still diagnostic of pliopithecoids, and some of crouzeliids and crouzeliines, as confirmed by our previous cladistic analysis.

Actually, this is not implied by the use of the past tense. The phrasing here is equivalent to “that are thought to be”.

“Will the work be of significance to the field and related fields? How does it compare to the established literature? If the work is not original, please provide relevant references.” It is significant to those working in the area of human evolution, ape evolution, and anthropoid evolution. It adds important evidence to a question that is plagued by missing data.

Response: We agree.

“Does the work support the conclusions and claims, or is additional evidence needed?” This is complicated. The cladistic analysis reported in the paper doesn’t really address the important question of where *Pliobates* fits within anthropoids. It essentially asks what kind of stem catarrhine it is. Its teeth suggest that it is a pliopithecoid, while its postcrania suggest that it may be a stem hominoid. But the cladistic analysis just looks at its teeth in a matrix with very few non-pliopithecoid taxa. Also, the data files supporting the analysis are flawed.

Supplementary table 4 is missing a lot of important text.

Response: As clarified above by the editor’s comments, the issue with the files was already fixed (we apologize for the inconvenience). As for the reviewer’s contention that “the paper doesn’t really address the important question of where *Pliobates* fits within anthropoids”, we strongly disagree as we clearly asserted that *Pliobates* is a crouzeliine pliopithecoid, and the pliopithecoid status was even noted in the title. The reviewer is implicitly assuming that providing a cladogram is the only valid way to demonstrate that a given taxon is what morphology shows it to be. This can sometimes be of use, and it can also be misleading, as it clearly was in Alba et al. (2015), as the crouzeliid synapomorphies shown by the newly reported teeth of *Pliobates* unambiguously indicate that it is a pliopithecoid. We are not arguing that dental features are not subject to homoplasy, but clearly, dental morphology is the most reliable source of information for deciphering the taxonomic affinities of extinct primates. In this case, *Pliobates* shows some features that have been only found among pliopithecoids (such as the distal arm of the pliopithecine triangle), whereas postcranial convergences have been previously noted between crown hominoids, atelids, and even other taxa such as lorisids. This is further discussed below in response to additional comments by the reviewer in this regard, but to make a long story short, in the revised version, we also performed a second cladistic analysis that includes a wider representation of taxa as well as cranial and postcranial features. Nevertheless, for the reasons explained below, we also analyzed separately two partitions of the data, as the postcranium is clearly more affected by functional than true phylogenetic signal. Regarding Supplementary Table 4 the file has been uploaded again (Excel file) as Supplementary Data 1.

Let me clarify what I was, and am, arguing. I am simply saying that there is a difference between a taxonomic assessment reached by thinking and a phylogenetic placement reached as the result of a cladistic analysis. When you see the lower teeth of *Pliobates*, you may notice some features

that in combination strike you as a distinctive set of traits seen only in pliopithecoids, and crouzeliids specifically. You could argue on that basis that it is indeed a member of that family. In so doing, you would arguably be using cladistic reasoning even without having conducted a computer-assisted cladistic analysis. You might feel that there is no need to conduct such a formal analysis just to test your hypothesis that *Pliobates* is a pliopithecoid, but I contest that ideally you should (while recognizing the burden of it). My reasoning is that not doing so assumes too much about a group of anthropoids about which there is still much confusion and many surprises still to come – like the ones supplied by *Pliobates*. To be clear again, I think a crouzeliid position for *Pliobates* is the best hypothesis at present, but I don't think we should completely rule out the possibility that we still have some of this wrong. We used to think that the entepicondylar foramen was as informative as the form of the ectotympanic among early catarrhines. *Pliobates* shows us that it is not. The ectotympanic is known for four out of the dozens of Miocene non-cercopithecoid catarrhine species. I'm not sure how long that picture will remain uncomplicated if we manage to find more. I realize that the authors didn't give any special weight to that character – they don't have to. The fact that it has a current distribution that is uncomplicated by apparent homoplasy gives it influence.

So my point is just this – what unfolded between *Aegyptopithecus* and *Pierolapithecus*, or even *Ekembo*, is still very uncertain. When a new piece of significant evidence like *Pliobates* emerges, it should be given the opportunity to shed as much light on this as possible, and this requires a comprehensive phylogenetic analysis. I am not surprised to hear that the authors had such an endeavor planned. I also see nothing wrong with writing a paper exploring the detailed systematics within pliopithecoids, but the two are not the same, and only one of them is potentially of broad interest REDACTED.

"Are there any flaws in the data analysis, interpretation and conclusions? Do these prohibit publication or require revision?" At minimum, Supp. Table 4 has to be fixed. It is missing chunks of information.

Response: As explained above, this was already fixed by the editor in the previous round. The data were reuploaded and correctly labeled this time.

Yes, this is resolved.

"Is the methodology sound? Does the work meet the expected standards in your field?" It is not sound if the conclusion it intends to reach is that *Pliobates* is a pliopithecoid.

Response: We strongly disagree with the reviewer's assessment in this regard, for the reasons anticipated above and explained in greater detail below. No cladistic analysis would be necessary to show this, and we already provided one that shows *Pliobates* deeply embedded within crouzeliid pliopithecoids based on the most reliable source of taxonomic information: teeth. Nevertheless, the broader cladistic analysis requested by the reviewer has been provided in the revised version.

See above.

"Is there enough detail provided in the methods for the work to be reproduced?" If Supp. Table 4 is completed.

Response: As noted above, this concern no longer applies.

Yes, this is resolved.

This manuscript presents valuable new information about *Pliobates*. We can now see that its teeth bear traits that suggest that it is a pliopithecoid. However, the cladistic analysis in the manuscript does not fully explore the possible phylogenetic implications of this because it is far too narrow both in terms of character data (being restricted to dental traits) and taxonomic representation. The analysis mostly speaks to the question of what its affinities may be within Pliopithecoidea if it is a pliopithecoid. This is largely irrelevant to the broader significance of the new data, which potentially speak to the question of whether or not *Pliobates* is a pliopithecoid, and consequently whether its ape-like postcranial traits represent convergent acquisitions with respect to crown hominoids. This is the question that makes the new fossils of great importance, and the cladistic analysis presented here doesn't advance that question at all. Unfortunately, the presentation of the analysis is also marred by what I assume are some formatting errors. The character table (Suppl Table 4) is missing chunks of text. This prevented me from examining the character support for various nodes.

Response: We appreciate that the reviewer understands the importance of the fossils described in our manuscript but respectfully disagree with their criticisms in the paragraph above. The newly described lower teeth do not merely "suggest" *Pliobates* is a pliopithecoid.

What, then, do cladistic analyses do? If we can make phylogenetic determinations without them, then what are they for? The newly described teeth don't even "suggest" anything – they do nothing, full stop. They simply are teeth. We, as scientists, can do things with the information we glean from them. Our perception of their morphology may lead us to hypothesize that they belong to a pliopithecoid. We should hypothesize that, and then test it by conducting a cladistic analysis that gives the OTU to which we assign them the opportunity to cluster with pliopithecoids or other anthropoids. That is what makes this a scientific endeavor. We don't just look at things and say what we think they are.

As noted in a previous response to this reviewer above, they clearly show this to be the case, to the extent that no cladistic analysis would be required to make the point that *Pliobates* is a pliopithecoid of the family Crouzeliidae. To say that our cladistic analysis "doesn't advance that question at all" is a most unfair criticism when it is taken into account that our cladistic results are the best-resolved phylogeny of pliopithecoids published so far and unambiguously recover *Pliobates* as a crouzeliine. The reviewer is missing the point that our cladistic analysis of dental traits (the only one provided in the previous version of the manuscript) was explicitly intended to clarify the phylogenetic relationships of *Pliobates* with other crouzeliids, not to confirm its pliopithecoid affinities.

This last sentence is entirely my point. The pliopithecoid status of *Pliobates* was not a result of a scientific analysis in this initial manuscript (n. b. I think it is now that they have expanded the analysis in revision).

In other words, the differential diagnosis would have been enough to demonstrate that it belongs to this pliopithecoid family, but our cladistic analysis was required to pinpoint to what crouzeliids it is more closely related.

But nobody reading Nature is going to care where precisely *Pliobates* falls within crouzeliids, so I was trying to encourage you to broaden the analysis so that it could speak to the broader question that actually might be of interest to a broader audience.

The reviewer implicitly questions the taxonomic assessment that *Pliobates* is a crouzeliid pliopithecoid and assumes that a cladistic analysis including also cranial and postcranial features, along with a broader representation of hominoids, might still indicate that *Pliobates* is a stem hominoid, as in Alba et al. (2015). Given the crouzeliid synapomorphies displayed by the lower teeth of *Pliobates* as well as the results of our dental-only cladistic analysis, we disagree

with such an interpretation, which entirely ignores the problem that *Pliobates* highlights regarding cladistic analysis, including postcranial data: namely, that there is a lot of postcranial homoplasy (at least, between *Pliobates* and crown hominoids) distorting the cladistic results. Probably, this would not happen if there were less missing data (for many fossil hominoid and pliopithecoid taxa). In any case, the bias introduced by postcranial homoplasy has been recently restated by Pugh (2022) JHE and even more explicitly by Urciuoli & Alba (2023) JHE, based on the contradictory results obtained by craniodental and postcranial data separately. As noted by the latter authors, a problem of long-branch attraction caused by postcranial convergences between hylobatids and hominids might be distorting the phylogenetic analyses of crown hominoids, and *Pliobates* indirectly supports the plausibility of this hypothesis by showing the distorting effect that such homoplasies had in Alba et al.'s (2015) cladogram. In other words, the postcranial features the reviewer wants us to include in the cladistic analysis are the main reason (coupled with the fact that lower teeth were previously unknown for *Pliobates*) why the previous cladistic analysis by Alba et al. (2015) recovered *Pliobates* as a hominoid.

And yet when you grudgingly did it anyway in the revision, didn't it end up nicely illustrating this point?

Other cladistic analyses, including additional taxa and using a different character selection (with less emphasis on these postcranial features), already recovered *Pliobates* as a pliopithecoid (Nengo et al., 2017 Nature; Gilbert et al., 2020 PRSB). Sincerely, we do not understand why the reviewer remains so skeptical and considers the pliopithecoid affinities of *Pliobates* to be unsettled after the more conclusive dental evidence provided in our manuscript. REDACTED

All my criticisms of this manuscript were intended to improve it. REDACTED

Said that, although we disagree with the validity of the reviewer's criticism in this regard, their request is not unreasonable.

If it's not unreasonable then why do you need to argue against it REDACTED

So, we decided to address it in the revision by performing a second cladistic analysis that includes a wider representation of taxa as well as craniodental and postcranial characters. This analysis was based on the original matrix by Alba et al. (2015), as this is the only one that has thus far recovered *Pliobates* as a hominoid, but updated based on the dental characters coded in our manuscript, coupled with some additional cranial features based on recently published literature. As a disclaimer, it should be noted that performing a total evidence (i.e., craniodental + postcranial) cladistic analysis of *Pliobates* was outside the scope of this paper. Therefore, until ongoing research is completed, for the cranial and postcranial characters, we had to mostly rely on the previous matrix of Alba et al. (2015), although with some improvements. We refrained from including species from the dental cladistic analysis that were not included in Alba et al. (2015), because their large number of missing data for cranial and postcranial features makes their position unstable and would collapse most of the clades. The cranial and postcranial characters included were updated according to the progress made in the field since 2015, for example, by replacing *Proconsul* s.l. by *Ekembo* (i.e., excluding *Proconsul* s.s.). We also grouped all *Pliopithecus* species (as in Alba et al., 2015) to perform the analysis at the genus level. However, unlike in the latter paper, we kept separate the dendropithecids *Simiolus* and *Dendropithecus*, as well as the crouzeliines *Crouzelia* and *Plesiopliopithecus*. Moreover, besides updating the dental characters based on the present paper, we added characters based on the inner ear semicircular canals and improved the coding of some other cranial and postcranial features based on recently published papers. Details on character modifications are given in Supplementary Methods of the revised version.

The results of the new cladistic analyses (Supplementary Fig. 6 in the revised version) not only support our previous assessment that *Pliobates* is a crouzeliid pliopithecoid but further confirm our suspicions about the confounding effect of postcranial homoplasies. As explained above, in all analyses considering (cranio)dental characters *Pliobates* branches off among other crouzeliids but, when postcranial characters are included, all pliopithecoids are recovered as a paraphyletic grouping of stem hominoids instead of stem catarrhine clade. In the postcranial-only analysis, all the clades are collapsed except crown hominoids (i.e., excluding *Ekembo*), and *Pliobates* is recovered as a stem hominoid. The reviewer is free to favor the results based on all evidence available, but considering the other analyses and especially the discrepancies between craniodental and postcranial analyses, our interpretation is that postcranial convergences between *Pliobates* and hominoids are misguiding the most parsimonious cladogram due to the wrong signal introduced by postcranial homoplasies. And, in any case, even if pliopithecoids were stem hominoids, this would not invalidate our assessment (as confirmed by the new cladistic analyses) that *Pliobates* is a crouzeliid.

So you did what I suggested, and the results nicely illustrate your point about postcranial homoplasy, which was what made this new information about *Pliobates* potentially interesting enough to be worthy of publication in a non-specialist journal. So why was I wrong to request it?

To clarify, I oppose the view that every paper about a fossil ape needs to include an exhaustive cladistic analysis. But the potential importance of the new data for *Pliobates* reported here cannot be realized without placing it in a more inclusive analysis. We need to give *Pliobates* (all of its anatomy) the opportunity to cluster with other anthropoids like crown hominoids with which it shares postcranial traits. This complex riddle will only be solved by including as much data as possible. The most critical limiting factor here is anatomical coverage for the fossil taxa, and that is what makes this discovery so very important. It just has to be put to better use. Otherwise, the present analysis could serve well as the basis for a JHE or AJBA paper about the potential phylogenetic placement of *Pliobates* within Pliopithecoidea. But the present analysis does not establish *Pliobates* as a pliopithecoid because it does not test the alternatives.

Response: The reviewer's assertion contradicts their request to provide a more comprehensive cladogram in the present manuscript.

No, I'm explaining why I think this paper needs one even though not every paper needs one.

In addition, the reviewer contends that the systematic position of *Pliobates* can only be solved by including as much data as possible. However, we consider that postcranial data (due to abundant homoplasy) is the main factor that misled the results of Alba et al. (2015). The abundant missing data for many taxa (a problem noted by the reviewer), coupled with abundant postcranial morphology, is arguably distorting the results. Nevertheless, as already noted in the previous response to this reviewer, we decided to give *Pliobates* the chance to cluster with other anthropoids, while at the same time analyzing the craniodental and postcranial features separately, and the results based on all evidence available support its crouzeliid status but challenge instead the stem catarrhine status of pliopithecoids as a whole.

We would like to stress that we do not consider the reviewer's request unreasonable (this is why we fulfilled it), we just disagree that such analysis was necessary to conclusively show that *Pliobates* is a pliopithecoid. Performing a more comprehensive analysis of *Pliobates* not restricted to dental characters is something that we had been planning for years. At first, we regretted to be forced to do so at this time, when there is plenty of ongoing research about *Pliobates* that will hopefully enable to code additional cranial and postcranial characters of this taxon. Nevertheless, we now realize this was worth the effort, because our new cladistic results more clearly evince the confounding effect of postcranial homoplasy without contradicting the pliopithecoid status of *Pliobates*.

Exactly.

We therefore think that, thanks to the reviewer's insistence, the revised version is much more solid in this regard.

REDACTED

Both the Results and Discussion have been reorganized and rewritten to an important extent to accommodate space for the new cladistic results. The Methods (Cladistic analysis section) now include a small paragraph on the new cladistic analyses. The Abstract and Introduction were also modified accordingly. As specified above, the Supplementary Methods provide details on how the new analyses were performed. The Supplementary Data 3 and 4 provided along with the revised version of the manuscript correspond to the new character definitions and the new matrix, respectively. Supplementary Text 4 discusses in greater detail the new results that are shown in Supplementary Fig. 6.

Below are some specific comments by manuscript line number that may be helpful.

37: If NE is an abbreviation of Northeast, it should be spelled out the first time it is used. **Response:** Done, but we see no reason to capitalize "northeast".

54: I think it is important to think about how this emphasis was manifested methodologically.

Response: We are not sure what the reviewer implies with this comment. There is no objective way to determine into how many discrete characters the morphology of a particular bone must be subdivided. There are more objective ways to discretize continuous features, but not to determine how many characters best represent a particular structure. This is indeed one of the main reasons as to why there is an inherent subjectivity in cladistic analyses, the characters coded by different authors will largely depend on their expertise and other unconscious biases, and even those suspected to be homoplastic should not be removed, as (from a cladistic epistemological viewpoint) homoplasy can only be determined a posteriori based on the most parsimonious topology. An analysis of all the evidence available is always preferable on epistemological terms, but the outcome will be determined by the number of characters included from different areas that, for whatever reason, provide a different signal than others (see discussion in Urciuoli & Alba, 2023 JHE). Given this caveat, we consider that the most suitable approach is to be as clear as possible in this regard by following the example of Pugh (2022), who separately analyzed craniodental and postcranial features to detect the inconsistent topologies recovered for some taxa. Our interpretation of the results has already been outlined in previous responses to this reviewer and is exposed in greater detail in the revised Discussion.

56: And of course if they are homoplastic then they are not synapomorphies. Maybe just call them traits of characteristics.

Response: We understand the reviewer's viewpoint but consider that is preferable to just add "purported" before "synapomorphies".

That looks good.

59: Wouldn't the most obvious approach be to investigate its phylogenetic position using an expanded dataset that approaches total evidence as much as possible? You could say that the previous lack of informative dental material for Pliobates crippled such an approach, but because of this new discovery it is now possible.

Response: We disagree, for the reasons explained above, namely: (1) an expanded dataset would preclude analyzing the internal phylogeny of pliopithecoids because many of them have a lot of missing data regarding cranial and postcranial features; (2) there are strong reasons to suspect that the possession of homoplastic postcranial features led Alba et al.'s (2015) cladistic

analysis to recover *Pliobates* as a hominoid. Nevertheless, as also explained above, the reviewer's request is reasonable, and hence we added a second cladistic analysis based on an extended dataset with additional taxa and characters, as requested by the reviewer. See above for an outline of the new cladistic results, which confirm the crouzeliid status of *Pliobates* and illustrate the misleading effects of postcranial convergences.

So again, I'm wrong but you did it anyway and now the paper is better.

273: The analysis could hardly have found pliopithecoids to be anything else. The cladistic analysis in this manuscript includes only one taxon that is universally considered a crown catarrhine. Ekembo is thought by many, perhaps most, to be a very primitive hominoid, but this is by no means unanimous. This is a very good analysis of pliopithecoid phylogenetics, but it is not capable of speaking to the broader issues that make *Pliobates* broadly interesting.

Response: We think that's not the case, for two good reasons. First, we see no reason as to why our dental cladistic analysis could not have recovered pliopithecoids as the sister taxon of dendropithecids instead of as a more basal lineage of stem catarrhines, or even as stem hominoids more closely related to *Ekembo*, if the character scoring for *Pliobates* and other taxa would have been different. Second, we are not sure who the reviewer has in mind when asserting that not everyone agrees that Ekembo is a stem hominoid REDACTED, but the question here is that Alba et al. (2015) not only recovered *Pliobates* as a stem hominoid, but also as more derived than *Ekembo*, which is contradicted by the cladistic analysis provided in the previous version of the manuscript. In any case, this criticism no longer applies after the new cladistic analysis included in the revised version.

304: Having a combination of primitive and derived traits (which is true of most taxa) does not cause this kind of problem – homoplasy does.

Response: The reviewer is misrepresenting our original sentence, which specified a "mosaic of primitive (stem catarrhine-like) and derived (hominoid-like) features, particularly in the postcranium". In other words, *Pliobates* displays some derived features of crown hominoids while lacking features derived of crown catarrhines. A stem hominoid would be expected to display a mixture of primitive and derived features compared with crown hominoids, but not compared to crown catarrhines as a whole. Of course, this is caused by homoplasy, but we believe this is just another (and more vague) way to put it and that our original sentence is clear enough to keep it—particularly given the aim of the sentence, which was to explain the discrepancy between different cladistic analyses depending on the emphasis put on the primitive features or on the hominoid-like ones. No changes were thus introduced in this particular sentence.

I think you could clarify this by saying "derived (crown hominoid-like) features". This is now on line 360.

368: and whatever the *Lomorupithecus* face is.

382: this wording is much clearer than that on about line 360.

380. I'm afraid the analysis really doesn't speak to this question.

Response: This comment is difficult to understand. The sentence alluded reads "Deciphering the phylogenetic relationships between pliopithecoids, crown catarrhines, and other Miocene catarrhines is not the aim of this paper". So, of course the analysis provided in the previous version of the manuscript did not speak about this question. We felt compelled to prevent people from thinking otherwise, but either the reviewer misread the sentence to mean just the opposite or we misinterpreted their comment as a criticism when indeed the reviewer was agreeing with our passage. In any case, after the addition of a new, more comprehensive

cladistic analysis, the revised manuscript does more directly tackle this issue, and hence we modified the sentence to make it more clear: "Deciphering the phylogenetic relationships among pliopithecoids, crown catarrhines, and other Miocene catarrhines is not the main aim of this work but the results of our cladistic analyses support the view that pliopithecoids (including Pliobates) are stem catarrhines and that previous results indicating a stem hominoid status are probably attributable to the independent acquisition in crouzeiids of postcranial ape-like features".

Trait #75 from Supplementary Table 6 isn't in Supplementary Table 4. It is listed as one of the state changes between Node 16 and node 14 in Supplementary Table 6.

In fact, something has gone very wrong with Supp. Table 4. Many of the characters are missing, and there are sentence fragments interspersed in various places.

Response: We apologize for the inconvenience but don't understand what happened exactly, as the files were correctly formatted (provided as Excel files) and the problem according to the Editor is that we incorrectly uploaded them as Supplementary Information instead of Supplementary Data. Anyway, our mistake was already fixed by the Editor during the previous round and, when submitting the revision, we ensured that the files were correctly uploaded as Supplementary Data (1 and 2, respectively) as Word files.

454: I suggest removing "kind of"

RESPONSE TO REVIEWERS

Ref. NCOMMS-23-22790A

Reviewer #1:

The authors have adequately addressed the issues that were raised in the reviews and I recommend that the revised version be accepted for publication.

Response: We thank the reviewer for his positive feedback and the minor edits to the present version.

A few minor wrinkles that need to be resolved prior to publication:

L33: Scholars working on hylobatids object to the use of the term "lesser ape" because of its pejorative connotation. It would be better to use hylobatids instead.

Response: The original use of "lesser" vs. "great" apes referred to their size differences without implying any pejorative connotations and in our opinion the two terms are still widely used in the literature with these meaning (the dictionary also defines "lesser" as "used in names of animals and plants which are smaller than similar kinds"). However, to avoid misunderstandings we rephrased as suggested by the reviewer.

L54: "in" should read "on".

Response: Fixed.

L125-126 + L221: "very mesially tapering" would be better as "mesially very tapering"

Response: Rephrased as suggested.

L232: Delete "entirely". Redundant.

Response: Deleted.

L348: "resolutive" is an odd legal term to use here. Rephrase?

Response: We agree, rephrased as "informative".

L381: Delete hyphen between "stem" and "catarrhine"

Response: Deleted.

L458: "true phylogenetic signal". It's problematic to invoke phylogenetic "truth" in regard to fossil taxa. Rephrase?

Response: We used this expression twice in the manuscript ("is introducing more 'noise' than true phylogenetic signal" and "may override true phylogenetic signal"). We consider it is not problematic in this context, as we were opposing true (i.e., correct) phylogenetic signal with noise (i.e., wrong phylogenetic signal). This is a theoretical concept and a search in Google Scholar shows that it is very frequently used in the literature. Nevertheless, we understand the reviewer's concern. It is an assumption of phylogenetic inference that there is a single (true) phylogeny for any given group of taxa but this true phylogeny is unknowable by definition. Therefore, to avoid misunderstandings, we deleted "true" in both instances, as it is indeed implicit in the expression "phylogenetic signal" in the context of these sentences.

L461: "along" should read "among"

Response: Changed as suggested.

L536: Spelling of "macinnesi"

Response: Fixed here and also elsewhere in the Supplementary Information.

Reviewer #2:

As for my previous comments to the manuscript, I think they have been improved well in this revised version. The photographs of comparative specimens added in the

supplementary figures are nice and useful both for general readers and specialists. I think this manuscript is worth to be published with a few minor corrections (see below).

Response: We thank the reviewer for their positive feedback and suggestions for improvement.

Supplementary Figure 5

Check the calculation for the upper molars. The index may have been calculated as length/breadth (%), instead of breadth/length. As the upper molars are buccolingually broader than mesiodistally long, the breadth/length indices should be over 100%.

Response: The reviewer is completely right, thank you for catching this. The index was recomputed for the three upper molars and the Supplementary figure 5 modified accordingly. Moreover we realized that in the caption we originally wrote “buccolingual index” instead of “breadth/length index”. This typo was corrected.

Supplementary Text 4.

“Alternatively, it may be argued that that the total evidence results are” >

“Alternatively, it may be argued that the total evidence results are....” (Remove the unnecessary “that”.) **Response:** Deleted.

Reviewer #3:

The authors have addressed my principal criticism by adding an additional cladistic analysis that demonstrates the important point they were trying to make about postcranial homoplasy. REDACTED

Response: We are glad to see that the reviewer is satisfied by the changes introduced in the previous revision. We also regret that the reviewer did not take well our previous response to their comments. We have the right to disagree from the reviewer’s opinion but recognize we might have expressed our views too boldly in some instances. However we believe we used a respectful tone and, in any case, it was not our intention to offend the reviewer by any means.

The authors agree that my requested broader cladistic analysis is “reasonable”, a word they chose, but only after repeatedly arguing that it is not necessary and that I’m only requesting it because I don’t believe their attribution of *Pliobates* to the crouzeliid family of pliopithecoids. There are three things I want to say about this.

The first is that I don’t see why they think it is a reasonable suggestion if my reasons for requesting it are wrong. Wouldn’t that make it literally unreasonable – without reason?

Response: We refer to our detailed explanations in the response to reviewers provided in the previous round. However, just to explicitly answer the reviewer’s question: we considered the request was reasonable but not well reasoned (i.e., justified based on the arguments provided by the reviewer). This is because the new cladistic analysis was not necessary to show *Pliobates* is a crouzeliid pliopithecoid, as apparently argued by the reviewer. However, it was a welcome addition to show the readers why Alba et al.’s (2015) cladistic analysis supported the wrong status of *Pliobates* as a stem hominoid. It is noteworthy that we were able to show this mainly because we decided to analyze postcranial characters separately from craniodental ones. This was our initiative, not something recommended by the reviewer.

The second is that I think their objections are unreasonable. What they seem to be saying is that they can tell this species is a crouzeliid just by looking at it (based on its diagnostic traits), and only need a cladistic analysis to tell them more precisely what kind of pliopithecoid it is. Imagine if I did that with *Oreopithecus* and a selection of

nanzapithecines along with Ekembo, Saadania, and Aegyptopithecus. I would find out just what kind of nanzapitheine Oreopithecus is. I don't think that would convince Reviewer #1 that it is a nanzapithecine (or technically that they are all oreopithecids). I could expand greatly on why I think their objections were misguided, but there seems little point because they took my suggestion anyway.

Response: What we argued in our previous response was that our original cladistic analysis, based exclusively on dental traits, was devised to determine the phylogenetic relationships of *Pliobates* among the Pliopithecoidea (once we had convincingly shown, based on the possession of derived features, that it belonged to this group). We indeed consider that it is possible to determine that this species is at least a pliopithecoid, if not specifically a crouzeliid, by “looking” at its teeth (especially the lower molars), because the traits it possesses are clearly diagnostic. The reviewer criticized our original cladistic analysis because they assumed that it was devised to demonstrate *Pliobates* is actually a pliopithecoid and not a hominoid (or something else), instead of more precisely determining its phylogenetic relationships with other pliopithecoids. We further argued that a cladistic analysis based on all available evidence would not further resolve the systematic position of *Pliobates* because, depending on the postcranial characters included, it might yield again an incorrect topology (due to convergences with hominoids). This is indeed what we confirmed in the previous revision. So, in fact, we were forced to go beyond the reviewer's suggestion, by not only adding a taxonomically and anatomically more comprehensive cladistic analysis, but also by analyzing postcranial features separately. In any case, we concur with the reviewer that the addition made our manuscript more robust and hence remain grateful for their request, despite all the disagreements involved.

The third is this – I think the paper is much better for having added the broader cladistic analysis. I think it makes one of their important points more clearly than before.

Response: We agree upon this point!

Regarding the novelty of the information provided in this manuscript, it is my opinion that most of what is important about *Pliobates* was already published in the 2015 Science paper. It is true that the conclusions of the two papers are quite different because that one placed *Pliobates* as a stem hominoid, but the fact is that many of us (primate paleontologists who study and have published papers about catarrhines including pliopithecoids) did not think that it was a hominoid then based on the teeth already known. True, they did not exhibit any diagnostic pliopithecoid traits, but they were very unlike the teeth of crown hominoids. In that light, the refinement of its phylogenetic position offered here is significant, but I don't think it justifies publication in a high impact journal again.

Response: We appreciate the opinions provided by the reviewer but disagree in multiple aspects. In the paragraph above, the reviewer depicts our current manuscript as a mere reanalysis of previously available evidence, without recognizing that the different conclusions reached therein strongly rely on the detailed description and analysis of all the previously unpublished specimens described herein. The reviewer vaguely refers to other scholars that apparently disagreed about the hominoid status of *Pliobates*. However, the fact is that only a few authors (Benefit & McCrossin, 2015; Nengo et al., 2017; Gilbert et al., 2020; Ji et al., 2022) expressed such views in print, suggesting that *Pliobates* might be a pliopithecoid or another kind of stem catarrhine. None of these authors specifically proposed (much less demonstrated) that *Pliobates* is a crouzeliid closely related to *Crouzelia* and *Plesiopliopithecus*. The two latter genera are known on the basis of very fragmentary remains and it was previously entirely unsuspected that they (or any other pliopithecoids, by the way) could display so many postcranial convergences with crown hominoids. We believe that this has far-reaching

implications for catarrhine evolution as a whole and, for this reason, we do think that our manuscript merits publication in Nature Communications.

REDACTED

Further comments can be found in the uploaded response to the authors' response to reviews.

Response: The reviewer provided counter-replies to some of the responses provided by us in the previous round to their original comments. In our opinion, they do not add anything relevant to the general comments of the reviewer that we responded above. Nevertheless, since we were asked by the Editor to provide a full response to the reviewer, we have copied them below and provided one-by-one responses. We have omitted the original comment by the reviewer, but included our original response (in red font), the reviewer's counter-reply (in green), and our new response (in blue).

Original response: This is a fair account of our contribution, except that “thought to be diagnostic” implies they are no longer diagnostic, which is not the case. Some of the

dental traits revealed by the new material are still diagnostic of pliopithecoids, and some of crouzeliids and crouzeliines, as confirmed by our previous cladistic analysis.

Reviewer's reply: Actually, this is not implied by the use of the past tense. The phrasing here is equivalent to “that are thought to be”.

Current response: Thanks for the clarification, the wording sounded a bit ambiguous to us. We agree upon this point, then, although our clarification that some traits are diagnostic not of pliopithecoids as a whole, but of crouzeliids and crouzeliines, still applies.

Original response: As clarified above by the editor's comments, the issue with the files was already fixed (we apologize for the inconvenience). As for the reviewer's contention that “the paper doesn't really address the important question of where *Pliobates* fits within anthropoids”, we strongly disagree as we clearly asserted that *Pliobates* is a crouzeliine pliopithecoid, and the pliopithecoid status was even noted in the title. The reviewer is implicitly assuming that providing a cladogram is the only valid way to demonstrate that a given taxon is what morphology shows it to be. This can sometimes be of use, and it can also be misleading, as it clearly was in Alba et al. (2015), as the crouzeliid synapomorphies shown by the newly reported teeth of *Pliobates* unambiguously indicate that it is a pliopithecoid. We are not arguing that dental features are not subject to homoplasy, but clearly, dental morphology is the most reliable source of information for deciphering the taxonomic affinities of extinct primates. In this case, *Pliobates* shows some features that have been only found among pliopithecoids (such as the distal arm of the pliopithecine triangle), whereas postcranial convergences have been previously noted between crown hominoids, atelids, and even other taxa such as lorids. This is further discussed below in response to additional comments by the reviewer in this regard, but to make a long story short, in the revised version, we also performed a second cladistic analysis that includes a wider representation of taxa as well as cranial and postcranial features. Nevertheless, for the reasons explained below, we also analyzed separately two partitions of the data, as the postcranium is clearly more affected by functional than true phylogenetic signal. Regarding Supplementary Table 4 the file has been uploaded again (Excel file) as Supplementary Data 1.

Reviewer: Let me clarify what I was, and am, arguing. I am simply saying that there is a difference between a taxonomic assessment reached by thinking and a phylogenetic placement reached as the result of a cladistic analysis. When you see the lower teeth of

Pliobates, you may notice some features that in combination strike you as a distinctive set of traits seen only in pliopithecoids, and crouzeliids specifically. You could argue on that basis that it is indeed a member of that family. In so doing, you would arguably be using cladistic reasoning even without having conducted a computer-assisted cladistic analysis. You might feel that there is no need to conduct such a formal analysis just to test your hypothesis that *Pliobates* is a pliopithecoid, but I contest that ideally you should (while recognizing the burden of it). My reasoning is that not doing so assumes too much about a group of anthropoids about which there is still much confusion and many surprises still to come – like the ones supplied by *Pliobates*. To be clear again, I think a crouzeliid position for *Pliobates* is the best hypothesis at present, but I don't think we should completely rule out the possibility that we still have some of this wrong. We used to think that the entepicondylar foramen was as informative as the form of the ectotympanic among early catarrhines. *Pliobates* shows us that it is not. The ectotympanic is known for four out of the dozens of Miocene non-cercopithecoid catarrhine species. I'm not sure how long that picture will remain uncomplicated if we

manage to find more. I realize that the authors didn't give any special weight to that character – they don't have to. The fact that it has a current distribution that is uncomplicated by apparent homoplasy gives it influence.

So my point is just this – what unfolded between *Aegyptopithecus* and *Pierolapithecus*, or even *Ekembo*, is still very uncertain. When a new piece of significant evidence like *Pliobates* emerges, it should be given the opportunity to shed as much light on this as possible, and this requires a comprehensive phylogenetic analysis. I am not surprised to hear that the authors had such an endeavor planned. I also see nothing wrong with writing a paper exploring the detailed systematics within pliopithecoids, but the two are not the same, and only one of them is potentially of broad interest REDACTED

Current response: We essentially agree with all the comments provided by the reviewer above except that the significance of the paper relies so heavily on the inclusion of the new cladistic analysis. We agree it made the paper more robust. On the other hand, we regret that we had to perform the cladistic analysis with basically the same data used by Alba et al. (2015). Multiple analysis of the postcranial morphology of *Pliobates* are currently underway and we would have preferred to wait until such analyses are completed because the dental analysis was enough to conclusively show the crouzeliid status of *Pliobates*. But the disagreement with the reviewer is more apparent than real, i.e., based on the comments above it is not about the most likely phylogenetic relationships of this taxon but rather on what needed to be included in the present manuscript. As we followed the reviewer's advice and are happy with the final result, there is no need to continue arguing about this.

Original response: We appreciate that the reviewer understands the importance of the fossils described in our manuscript but respectfully disagree with their criticisms in the paragraph above. The newly described lower teeth do not merely “suggest” *Pliobates* is a pliopithecoid.

Reviewer: What, then, do cladistic analyses do? If we can make phylogenetic determinations without them, then what are they for? The newly described teeth don't even “suggest” anything – they do nothing, full stop. They simply are teeth. We, as scientists, can do things with the information we glean from them. Our perception of their morphology may lead us to hypothesize that they belong to a pliopithecoid. We should hypothesize that, and then test it by conducting a cladistic analysis that gives the OTU to which we assign them the opportunity to cluster with pliopithecoids or other anthropoids. That is what makes this a scientific endeavor. We don't just look at things and say what we think they are.

Current response: The reviewer is trying to justify their insistence that we needed to include a cladistic analysis while apparently forgetting that we did include a cladistic analysis (based exclusively on teeth) in the original version. The reviewer further implies that any taxonomic opinion must be necessarily based on a cladistic analysis, which is certainly not the case. There are different ways to assess morphological evidence, both qualitatively and quantitatively, from a taxonomic viewpoint and within an evolutionary framework, and they are all valid and useful. Cladistic analysis is devised to test among competing phylogenetic hypotheses but not always necessary to determine to what genus or family a particular species belong (of course, based on available knowledge that includes previous cladistic analyses). Anyway, we did perform a cladistic analysis based on teeth in the original version and included the more comprehensive cladistic analysis requested by the reviewer in the previous version. So, there seems to be no need to continue arguing about this.

Original response: As noted in a previous response to this reviewer above, they clearly show this to be the case, to the extent that no cladistic analysis would be required to make the point that *Pliobates* is a pliopithecoid of the family Crouzeliidae. To say that our cladistic analysis “doesn’t advance that question at all” is a most unfair criticism when it is taken into account that our cladistic results are the best-resolved phylogeny of pliopithecoids published so far and unambiguously recover *Pliobates* as a crouzeliine. The reviewer is missing the point that our cladistic analysis of dental traits (the only one provided in the previous version of the manuscript) was explicitly intended to clarify the phylogenetic relationships of *Pliobates* with other crouzeliids, not to confirm its pliopithecoid affinities.

Reviewer: This last sentence is entirely my point. The pliopithecoid status of *Pliobates* was not a result of a scientific analysis in this initial manuscript (n. b. I think it is now that they have expanded the analysis in revision).

Current response: The reviewer is literally saying that taxonomic assessments, even if based on detailed descriptions and comparisons, are unscientific unless backed up by a cladistic analysis. With all due respect, this assertion is untenable and appears an ad hoc argument to discredit our work. Besides, the original (dental-based) cladistic analysis included non-pliopithecoid taxa, including the stem hominoid *Ekembo*. The fact that *Pliobates* clustered well within crouzeliids instead of with *Ekembo* clearly supported the pliopithecoid status of the former. Why the reviewer disregards this as unscientific is beyond our understanding.

Original response: In other words, the differential diagnosis would have been enough to demonstrate that it belongs to this pliopithecoid family, but our cladistic analysis was required to pinpoint to what crouzeliids it is more closely related.

Reviewer: But nobody reading Nature is going to care where precisely *Pliobates* falls within crouzeliids, so I was trying to encourage you to broaden the analysis so that it could speak to the broader question that actually might be of interest to a broader audience.

Current response: The reviewer is missing the point that, if *Pliobates* is a crouzeliid, then by definition it is a pliopithecoid. By the way, we submitted our manuscript to Nature Communications, not to Nature, and we already explored in the original version the broader implications of this find. Illustrating with the new cladistic analysis that previous results supporting its stem hominoid status were misguided by postcranial convergences is a welcome addition, but the point was already made in the original version and supported by cladistic evidence.

Original response: The reviewer implicitly questions the taxonomic assessment that *Pliobates* is a crouzeliid pliopithecoid and assumes that a cladistic analysis including

also cranial and postcranial features, along with a broader representation of hominoids, might still indicate that *Pliobates* is a stem hominoid, as in Alba et al. (2015). Given the crouzeioid synapomorphies displayed by the lower teeth of *Pliobates* as well as the results of our dental-only cladistic analysis, we disagree with such an interpretation, which entirely ignores the problem that *Pliobates* highlights regarding cladistic analysis, including postcranial data: namely, that there is a lot of postcranial homoplasy (at least, between *Pliobates* and crown hominoids) distorting the cladistic results. Probably, this would not happen if there were less missing data (for many fossil hominoid and pliopithecoid taxa). In any case, the bias introduced by postcranial homoplasy has been recently restated by Pugh (2022) JHE and even more explicitly by Urciuoli & Alba (2023) JHE, based on the contradictory results obtained by craniodental and postcranial data separately. As noted by the latter authors, a problem of long-branch attraction caused by postcranial convergences between hylobatids and hominids might be distorting the phylogenetic analyses of crown hominoids, and *Pliobates* indirectly supports the plausibility of this hypothesis by showing the distorting effect that such homoplasies had in Alba et al.'s (2015) cladogram. In other words, the postcranial features the reviewer wants us to include in the cladistic analysis are the main reason (coupled with the fact that lower teeth were previously unknown for *Pliobates*) why the previous cladistic analysis by Alba et al. (2015) recovered *Pliobates* as a hominoid.

Reviewer: And yet when you grudgingly did it anyway in the revision, didn't it end up nicely illustrating this point?

Current response: This is only the case because we chose to analyze postcranial features separately from craniodental ones, so as to be able to illustrate the misleading effect of postcranial convergences. Indeed, without our original dental-only analysis and detailed assessment of dental morphology, and just following the reviewer's suggestion to analyze all available information altogether, we might have ended up concluding that not only *Pliobates*, but all pliopithecoids, are stem hominoids. So, we thank the reviewer for the encouragement but the new analyses only nicely illustrated our point because we went beyond the reviewer's suggestion.

Original response: Other cladistic analyses, including additional taxa and using a different character selection (with less emphasis on these postcranial features), already recovered *Pliobates* as a pliopithecoid (Nengo et al., 2017 Nature; Gilbert et al., 2020 PRSB). Sincerely, we do not understand why the reviewer remains so skeptical and considers the pliopithecoid affinities of *Pliobates* to be unsettled after the more conclusive dental evidence provided in our manuscript. REDACTED

Reviewer: All my criticisms of this manuscript were intended to improve it.
REDACTED

Current response: We sincerely appreciate the efforts put by the reviewer in reviewing our manuscript. This was not explicitly noted in the acknowledgments of the revision just because the journal does not allow to do so. Besides, we reiterate that we did not aim to offend the reviewer (and still do not). REDACTED

Original response: Said that, although we disagree with the validity of the reviewer's criticism in this regard, their request is not unreasonable.

Reviewer: If it's not unreasonable then why do you need to argue against it
REDACTED

Current response: This has already been responded above. We regret that the reviewer felt attacked by our comment, as this was not our intention.

Original response: So, we decided to address it in the revision by performing a second cladistic analysis that includes a wider representation of taxa as well as craniodental and postcranial characters. This analysis was based on the original matrix by Alba et al. (2015), as this is the only one that has thus far recovered *Pliobates* as a hominoid, but updated based on the dental characters coded in our manuscript, coupled with some additional cranial features based on recently published literature. As a disclaimer, it should be noted that performing a total evidence (i.e., craniodental + postcranial) cladistic analysis of *Pliobates* was outside the scope of this paper. Therefore, until ongoing research is completed, for the cranial and postcranial characters, we had to mostly rely on the previous matrix of Alba et al. (2015), although with some improvements. We refrained from including species from the dental cladistic analysis that were not included in Alba et al. (2015), because their large number of missing data for cranial and postcranial features makes their position unstable and would collapse most of the clades. The cranial and postcranial characters included were updated according to the progress made in the field since 2015, for example, by replacing *Proconsul* s.l. by *Ekembo* (i.e., excluding *Proconsul* s.s.). We also grouped all *Pliopithecus* species (as in Alba et al., 2015) to perform the analysis at the genus level. However, unlike in the latter paper, we kept separate the dendropithecids *Simiolus* and *Dendropithecus*, as well as the crouzeliines *Crouzelia* and *Plesiopliopithecus*. Moreover, besides updating the dental characters based on the present paper, we added characters based on the inner ear semicircular canals and improved the coding of some other cranial and postcranial features based on recently published papers. Details on character modifications are given in Supplementary Methods of the revised version. The results of the new cladistic analyses (Supplementary Fig. 6 in the revised version) not only support our previous assessment that *Pliobates* is a crouzeliid pliopithecoid but further confirm our suspicions about the confounding effect of postcranial homoplasies. As explained above, in all analyses considering (cranio)dental characters *Pliobates* branches off among other crouzeliids but, when postcranial characters are included, all pliopithecoids are recovered as a paraphyletic grouping of stem hominoids instead of stem catarrhine clade. In the postcranialonly analysis, all the clades are collapsed except crown hominoids (i.e., excluding *Ekembo*), and *Pliobates* is recovered as a stem hominoid. The reviewer is free to favor the results based on all evidence available, but considering the other analyses and especially the discrepancies between craniodental and postcranial analyses, our interpretation is that postcranial convergences between *Pliobates* and hominoids are misleading the most parsimonious cladogram due to the wrong signal introduced by postcranial homoplasies. And, in any case, even if pliopithecoids were stem hominoids, this would not invalidate our assessment (as confirmed by the new cladistic analyses) that *Pliobates* is a crouzeliid.

Reviewer: So you did what I suggested, and the results nicely illustrate your point about postcranial homoplasy, which was what made this new information about *Pliobates* potentially interesting enough to be worthy of publication in a non-specialist journal. So why was I wrong to request it?

Current response: We respectfully disagree because what we did is not just what the reviewer requested, we went well beyond that and, thanks to this, we could illustrate our point. We have already responded to this above and still believe that the additional cladistic analyses, even if a welcome addition, were not necessary to make the paper worthy of publication in Nature Communications, as its broader implications were already stated in the original version. Yet we agree that the final result is more robust and better articulated, and reiterate our thanks to the reviewer for their suggestions and criticisms, which undoubtedly helped us to improve the paper. This delayed several

months the completion of the PhD by the first author of this paper but we're satisfied with the end result, and this is all that should matter.

Original response: The reviewer's assertion contradicts their request to provide a more comprehensive cladogram in the present manuscript.

Reviewer: No, I'm explaining why I think this paper needs one even though not every paper needs one.

Current response: The original paper already had a cladistic analysis that could have potentially favored a hominoid instead of pliopithecoid status for *Pliobates*, so we respectfully disagree with the reviewer's justification of his request. We will not agree upon this point, which is no longer relevant, as we provided the requested cladistic analysis.

Original response: We therefore think that, thanks to the reviewer's insistence, the revised version is much more solid in this regard.

Reviewer: Then perhaps you could have edited your responses REDACTED

Current response: We disagree about the qualifiers used by the reviewer and think that our tone was assertive but respectful. We are sorry the reviewer thinks otherwise but it is our right as authors to respectfully disagree with the reviewers as long as we provide justified rebuttals—which we did.

Original response: We disagree, for the reasons explained above, namely: (1) an expanded dataset would preclude analyzing the internal phylogeny of pliopithecoids because many of them have a lot of missing data regarding cranial and postcranial features; (2) there are strong reasons to suspect that the possession of homoplastic postcranial features led Alba et al.'s (2015) cladistic analysis to recover *Pliobates* as a hominoid. Nevertheless, as also explained above, the reviewer's request is reasonable, and hence we added a second cladistic analysis based on an extended dataset with additional taxa and characters, as requested by the reviewer. See above for an outline of the new cladistic results, which confirm the crouzeliid status of *Pliobates* and illustrate the misleading effects of postcranial convergences.

Reviewer: So again, I'm wrong but you did it anyway and now the paper is better.

Current response: Yes, basically this is our opinion. Nevertheless, we reiterate our thanks to the reviewer.

Original response: The reviewer is misrepresenting our original sentence, which specified a “mosaic of primitive (stem catarrhine-like) and derived (hominoid-like) features, particularly in the postcranium”. In other words, *Pliobates* displays some derived features of crown hominoids while lacking features derived of crown catarrhines. A stem hominoid would be expected to display a mixture of primitive and derived features compared with crown hominoids, but not compared to crown catarrhines as a whole. Of course, this is caused by homoplasy, but we believe this is just another (and more vague) way to put it and that our original sentence is clear enough to keep it—particularly given the aim of the sentence, which was to explain the discrepancy between different cladistic analyses depending on the emphasis put on the primitive features or on the hominoid-like ones. No changes were thus introduced in this particular sentence.

Reviewer: I think you could clarify this by saying “derived (crown hominoid-like) features”. This is now on line 360.

Current response: We agree. We added “crown” before “hominoid-like”.

Reviewer: 368: and whatever the *Lomorupithecus* face is.

Current response: The reviewer refers to our sentence “The cranial morphology of *Pliobates* is also ambiguous because some similarities with hylobatids (anteriorly situated orbits, broad interorbital distance, short face, low zygomatic roots) are also displayed by the pliopithecoid *Epipliopithecus*²⁷, stem hominoids such as nyanzapithecids², and the dendropithecoid *Micropithecus*²⁸.”. The reviewer refers to a genus that was originally described as an African pliopithecoid but subsequently reinterpreted as a likely stem catarrhine from East Africa, maybe related to dendropithecids. While the reviewer is right, the cranial remains of *Lomorupithecus* are quite fragmentary and we do not consider it necessary to mention it explicitly. We however rephrased the end of the sentence (“small-bodied catarrhines from East Africa such as the dendropithecoid *Micropithecus*”) to make it clear that *Micropithecus* is not the only one showing such features.

382: this wording is much clearer than that on about line 360.

Current response: The reviewer refers to our sentence “This is most clearly evidenced by the postcranium of *Pliobates*, which reveals a mosaic of stem catarrhine-like features combined with crown catarrhine and even crown hominoid synapomorphies absent in *Epipliopithecus*¹”. This sentence is probably clearer because it specifically refers to postcranial features and provides additional detail, while the previous one was not restricted to the postcranium. The previous sentence has already been slightly modified following the reviewer’s request (see above), so we think that no further changes are required.

Reviewer: 454: I suggest removing “kind of”

Current response: Deleted.